# Identifying small-molecules binding sites in RNA conformational ensembles with SHAMAN

F. P. Panei[1,2,3], P. Gkeka [1] ✉ & M. Bonomi [2] ✉

The rational targeting of RNA with small molecules is hampered by our still limited understanding of RNA structural and dynamic properties. Most in silico tools for binding site identification rely on static structures and therefore cannot face the challenges posed by the dynamic nature of RNA molecules. Here, we present SHAMAN, a computational technique to identify potential small-molecule binding sites in RNA structural ensembles. SHAMAN enables exploring the conformational landscape of RNA with atomistic molecular dynamics simulations and at the same time identifying RNA pockets in an efficient way with the aid of probes and enhanced-sampling techniques. In our benchmark composed of large, structured riboswitches as well as small, flexible viral RNAs, SHAMAN successfully identifies all the experimentally resolved pockets and ranks them among the most favorite probe hotspots. Overall, SHAMAN sets a solid foundation for future drug design efforts targeting RNA with small molecules, effectively addressing the long-standing challenges in the field.

RNA molecules, initially thought to be only carriers of genetic information from gene to proteins, are now known to perform a variety of biological functions, such as regulating the process of protein synthesis and defending against the entry of foreign nucleic acids into cells[1–4]. Alongside these findings, modulation of RNA functions is becoming a promising therapeutic approach for treating diseases such as cancer, viral infections, cardiovascular and muscular disorders, and neurodegenerative conditions[5–7]. Besides classical approaches, such as the design of antisense oligonucleotides interfering with mRNAs or directly editing RNA with CRISPR-Cas9, targeting RNA with small molecules is emerging as a promising strategy[8–11] in terms of number of potential targets, bioavailability, and delivery[11–15]. Although in recent years the research in this field has surged[16,17], the number of FDA-approved drugs is still limited and the compounds currently available on the market were identified exclusively by costly and time-consuming experimental screenings[16–18].

Computer-aided drug design (CADD) provides several essential tools to assist various stages of drug discovery, from druggability assessment to virtual screening for hit identification, binding affinity calculations, and generative methods for lead optimization. While these tools are well established for proteins, their application to RNA molecules is still in its infancy. The available biochemical and structural data is gradually elucidating the chemical properties of RNA binders[19] and the structural properties of RNA binding sites[20]. This knowledge has been stimulating the development of ligand-[21,22] and 2D structure-[23–25] based virtual screening approaches, 3D binding-site detection tools[26–30], docking software[31–34] and scoring functions[35–38] specific for RNA molecules. However, our understanding of the structural and dynamic properties of RNA molecules and their interaction with small molecules still remains limited, thus ultimately hindering the rational design of novel and effective compounds[39].

In the cellular context, function-specific biological signals trigger complex multi-step RNA conformational changes that in turn guide a variety of RNA functions, such as ligand sensing and signaling, catalysis, or co-transcriptional folding[40,41]. These conformational changes and the underlying dynamics are influenced both by the inherent

[1]Integrated Drug Discovery, Molecular Design Sciences, Sanofi, Vitry-sur-Seine, France. [2]Institut Pasteur, Université Paris Cité, CNRS UMR 3528, Computational Structural Biology Unit, Paris, France. [3]Sorbonne Université, Ecole Doctorale Complexité du Vivant, Paris, France. ✉e-mail: Paraskevi.Gkeka@sanofi.com; mbonomi@pasteur.fr

flexibility of RNA molecules, i.e., many large-scale motional modes spanning a variety of timescales, and other cellular co-factors[42]. Despite the significant efforts to characterize RNA dynamics using both experimental[43], in-silico[44], and integrative approaches[45], most available tools for CADD, and in particular for the identification of small molecules binding sites, still rely on a static description of RNA structure[26–30]. The only exception is SILCS-RNA[29] where potential binding sites are identified by exploring the conformation of the target RNA with small cosolvent probes, similar to mixed-solvent approaches already extensively used for proteins[46]. While SILCS-RNA can describe small structural rearrangements induced by the probes, it is not designed to capture large RNA conformational changes and, therefore, it is not able to detect binding sites present in metastable states that are marginally populated yet crucial for therapeutic applications[39–41,47].

Here, we present SHAdow Mixed solvent metAdyNamics (SHA-MAN), a computational technique for binding site identification in dynamic RNA structural ensembles. Thanks to its unique parallel architecture, SHAMAN allows at the same time to: (i) explore the conformational landscape of RNA with atomistic explicit-solvent molecular dynamics (MD) simulations driven by state-of-the-art forcefields and (ii) identify potential small-molecules binding sites in an efficient way with the aid of probes and the metadynamics[48] enhanced-sampling technique. SHAMAN was benchmarked on a set of biologically relevant target systems, including large, structured riboswitches as well as smaller highly dynamic RNAs involved in viral proliferation. Our method successfully identified all the experimentally resolved pockets present in our benchmark set and was able to rank them among the most favorite probe hotspots. Our work constitutes an advanced computational pipeline for binding site identification in dynamic RNA structural ensembles, thus providing crucial information for structure-based rational design of novel compounds targeting RNA.

## Results

This section is organized as follows. First, we provide a general overview of SHAMAN and illustrate its accuracy in identifying experimentally resolved binding sites in a set of biologically relevant RNA targets. Second, we focus on the probes used in our SHAMAN simulations and investigate their relation to physico-chemical features of both the RNA pockets and the small molecules bound to them in known experimental structures. We then compare SHAMAN with state-of-the-art tools for binding site prediction in RNA. Finally, we present two case studies, the FNM riboswitch and the HIV-1 TAR, to (i) demonstrate how SHAMAN can be used to study well-structured as well as more flexible RNAs; (ii) highlight the main strengths of our technique in modeling both local and global flexibility of the target. A complete analysis of the systems in our benchmark set is reported in Supplementary Information (Supplementary Analysis and Figs. S8–S13).

### Overview of the SHAMAN approach

SHAMAN is a computational technique that uses small fragments or probes and atomistic explicit-solvent MD simulations to identify potential small-molecule binding sites in RNA structural ensembles (Fig. 1A). SHAMAN is based on a unique architecture in which multiple replicas of the system are simulated in parallel (Fig. 1B). A mother simulation, containing only RNA and possibly structural ions, explores the conformational landscape of the target and communicates the positions of the RNA atoms to the replicas. Each replica contains a different probe that explores the RNA conformation provided by the mother simulation using the metadynamics enhanced-sampling approach[48]. Soft positional restraints applied to the RNA backbone atoms of the replica allow for local induce-fit effects caused by the probes, while following or shadowing the conformational changes of the mother RNA simulation. This parallel architecture enables an efficient exploration of the same RNA conformation by different probes

and the identification, for each representative cluster of RNA conformations, of a set of potential small-molecule binding sites or SHA-MAPs (Fig. 1C). Each SHAMAP corresponds to a region of space occupied with high probability by at least one probe and is ranked by the binding free energy $\Delta G$ of the probe(s) to a specific RNA conformation (Fig. 1D). A more detailed description of SHAMAN is provided in Methods.

### Benchmark of the SHAMAN accuracy

The accuracy of SHAMAN in identifying experimentally resolved binding sites was evaluated on 7 biologically relevant systems, including riboswitches (Fig. 2A) and viral RNAs (Fig. 2B). For each system, SHAMAN simulations were initialized from both holo conformations after the removal of the ligand (holo-like) and, when available, apo conformations, resulting in a total of 12 runs (Tab. S1 and S2). The validation set was composed of 14 unique binding pockets obtained from 69 experimental structures of riboswitches (Tab. S3) and viral RNAs (Tab. S4) in complex with different ligands. For each simulation, the accuracy was defined in terms of the distance between our SHA-MAPs and the ligand position in the reference experimental structures (Eq. 10 and Fig. 2C).

SHAMAN was able to identify the experimentally resolved pockets in all the systems of our benchmark set, both when initializing the simulations from holo-like and apo conformations (Tab. S5 and S6). Most importantly, the experimental binding sites were ranked among the most probable SHAMAPs in each corresponding run. To quantify the rank, we defined the difference in binding free energy $\Delta\Delta G$ between each SHAMAP and the one with lowest free energy (Eq. 9). When starting from the apo conformation of the target RNA, the $\Delta\Delta G$ of the SHAMAPs overlapping with the ligands was in 80% of cases below $k_B T$ and in the 100% of cases below $2k_B T$ (Fig. 2D). When starting from holo-like conformations, these percentages dropped to 64% and 84% (Fig. 2D). Ranking the experimental binding pockets among the SHAMAPs with lowest free energy (top scored) is fundamental in the context of CADD, and in particular in virtual screening applications (Discussion).

The geometrical proximity of our SHAMAPs to the experimental binding sites present in our benchmark set was noteworthy. The average distance between the centers of the interacting sites overlapping with a ligand and its position in the experimental structure was equal to 3.8 Å and 4.4 Å in the holo-like (Fig. 2E, upper panel) and apo (Fig. 2E, lower panel) cases, respectively. Both values are relatively small when compared to the distance threshold used in our validation criterion (Eq. 10), which was defined as the sum of the radius of gyration of the SHAMAPs (on average ~1.6 Å, Fig. S1A) and the ligand (on average ~3.7 Å, Fig. S1B). As expected, this proximity to the experimental binding sites was remarkably greater in the simulations initiated from holo-like conformations in which the binding sites were already present. As a matter of fact, 22% of the successful interacting sites identified in the holo-like simulations were close to the experimental pocket by half of our distance threshold, while this holds only for 1% of the apo simulations.

### Analysis of the probes

Two sets of probes were used in the SHAMAN benchmark described in the previous section. The first set of 8 probes (Tab. S7) was previously used in the development of SILCS-RNA[29] and was mostly composed of compounds selected to represent specific types of interaction with the RNA target. This set includes: acetate (ACEY), benzene (BENX), dimethyl-ether (DMEE), formamide (FORM), imidazole (IMIA), methyl-ammonium (MAMY), methanol (MEOH), and propane (PRPX). A second set of 5 probes (Tab. S8) was generated in this work using a fragmentation protocol (Methods) applied to ligands present in (i) the HARIBOSS[20] database of RNA-ligand resolved structures (https://hariboss.pasteur.cloud); and (ii) the R-BIND[24] database of bioactive

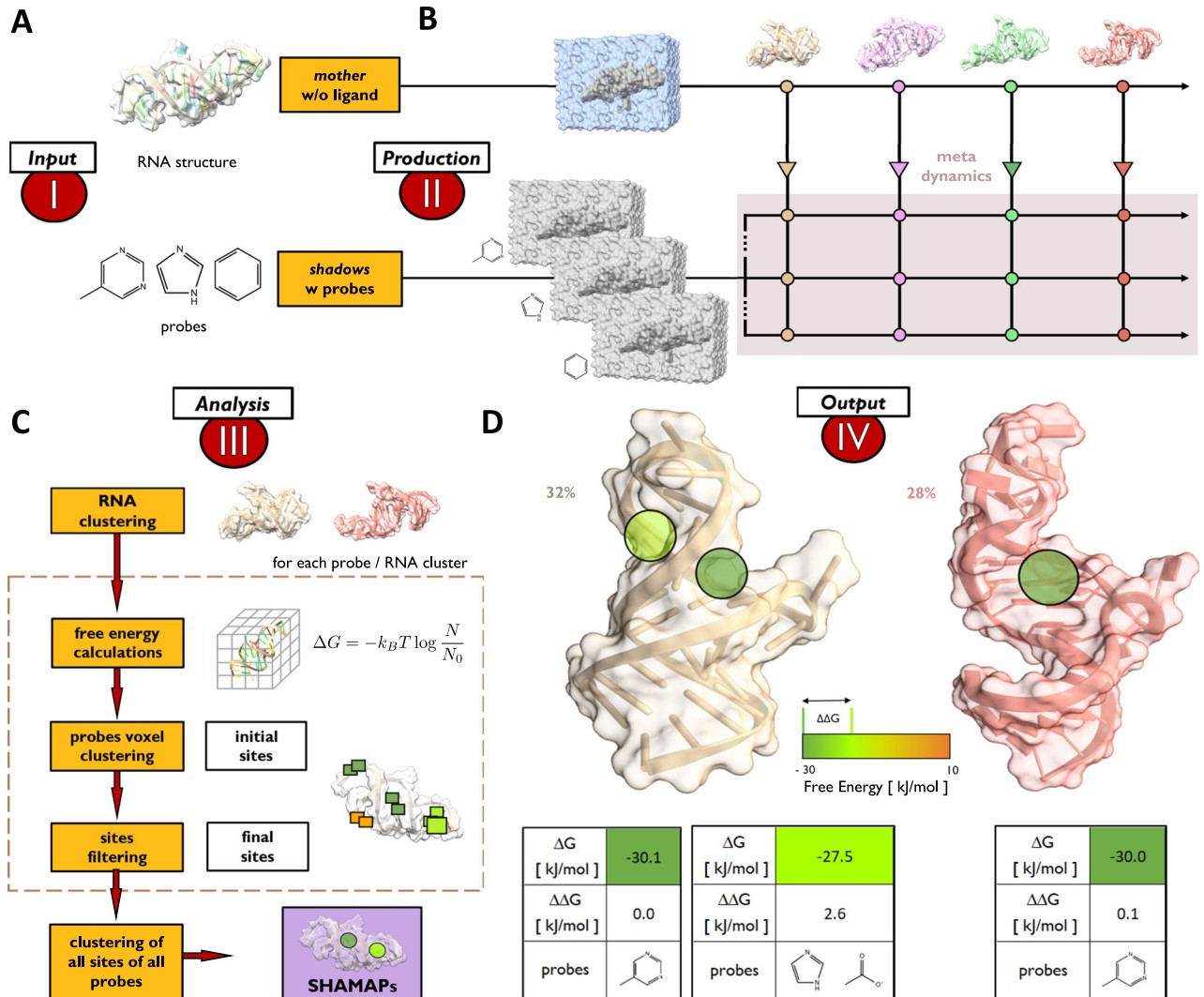

**Fig. 1 | Overview of the SHAMAN approach. A** Input stage: Selection of the RNA target structure and of the probes to initialize the mother and replica systems, each one with a different probe. **B** Production stage: the unbiased/unrestrained MD simulation of the mother system communicates the positions of the RNA backbone atoms to the replicas, which are restrained to follow the mother like shadows. The probe exploration of the RNA conformation is accelerated by metadynamics. **C** Analysis stage (from top to bottom): (i) the sampled RNA ensemble is clustered into a set of representative conformations; (ii) for each cluster and probe, a free-energy map is calculated from the probe occupancy during the course of the simulation; (iii) voxels in the free-energy maps are clustered together into interacting sites; (iv) for each interacting site, free energy and buriedness score are calculated and sites too exposed to solvent are discarded; (v) for each RNA cluster, all interacting sites obtained from all probes are clustered together into SHAMAPs. **D** Output stage: two RNA representative clusters with population equal to 32% (light brown, left panel) and 28% (pink, right panel) with the corresponding SHA-MAPs (green circles). For each SHAMAP, we provide the binding free energy to RNA ($\Delta G$) and the difference with respect to the lowest free energy (top scored).

small molecules targeting RNA (https://rbind.chem.duke.edu). This second set includes mostly aromatic compounds: benzene (BENX), dihydro-pyrido-pyrimidinone-Imidazo-pyridine (BENF), benzothiophene (BETH), methyl-pyrimidine (MEPY), and the cyclic non-aromatic piperazine (PIRZ).

We first explored the relation between the probes that successfully identified experimental binding sites and some of the structural features of RNA pockets. Aromatic probes showed a preference for exploring cavities buried deep inside the RNA structure (Fig. 3A, dark green bars), with an estimated average buriedness of $0.75 \pm 0.06$, which is relatively high compared to known RNA-small molecules pockets (Fig. 3B). On the other hand, non-aromatic probes displayed two distinct patterns. FORM, MEOH, and MAMY selectively explored shallow pockets with an average buriedness of $0.59 \pm 0.04$ (Fig. 3A, olive green bars), while DMEE, PRPX and ACEY promiscuously explored pockets with varying solvent exposure and an average buriedness of $0.70 \pm 0.08$ (Fig. 3A). PIRZ exhibited an intermediate

behavior, with an average buriedness of $0.65 \pm 0.06$ (Fig. 3A, brown bar). Aromatic probes were particularly successful (66% of cases) in identifying riboswitches binding sites, which in our validation set typically resided in buried cavities (Fig. 3C). For example, the location of the representative riboswitch binder GNG (PDB 3ski) was exclusively identified by aromatic probes (Fig. 3D). On the other hand, aliphatic probes identified pockets with high likelihood (70%) in viral RNAs (Fig. 3E), whose inherent flexibility resulted in shallow cavities exposed to solvent. An example is the binding site of SS0, a typical viral RNA binder (PDB 3tzr), which was identified primarily by non-aromatic probes (Fig. 3F).

Although the main goal of SHAMAN is pocket identification, motivated by its perspective use in virtual screening and ligand optimization (Discussion) we also investigated the link between the similarity of a given probe to a ligand and its ability to identify the corresponding experimental pocket. We started by comparing standard physico-chemical properties of the entire ligand or the

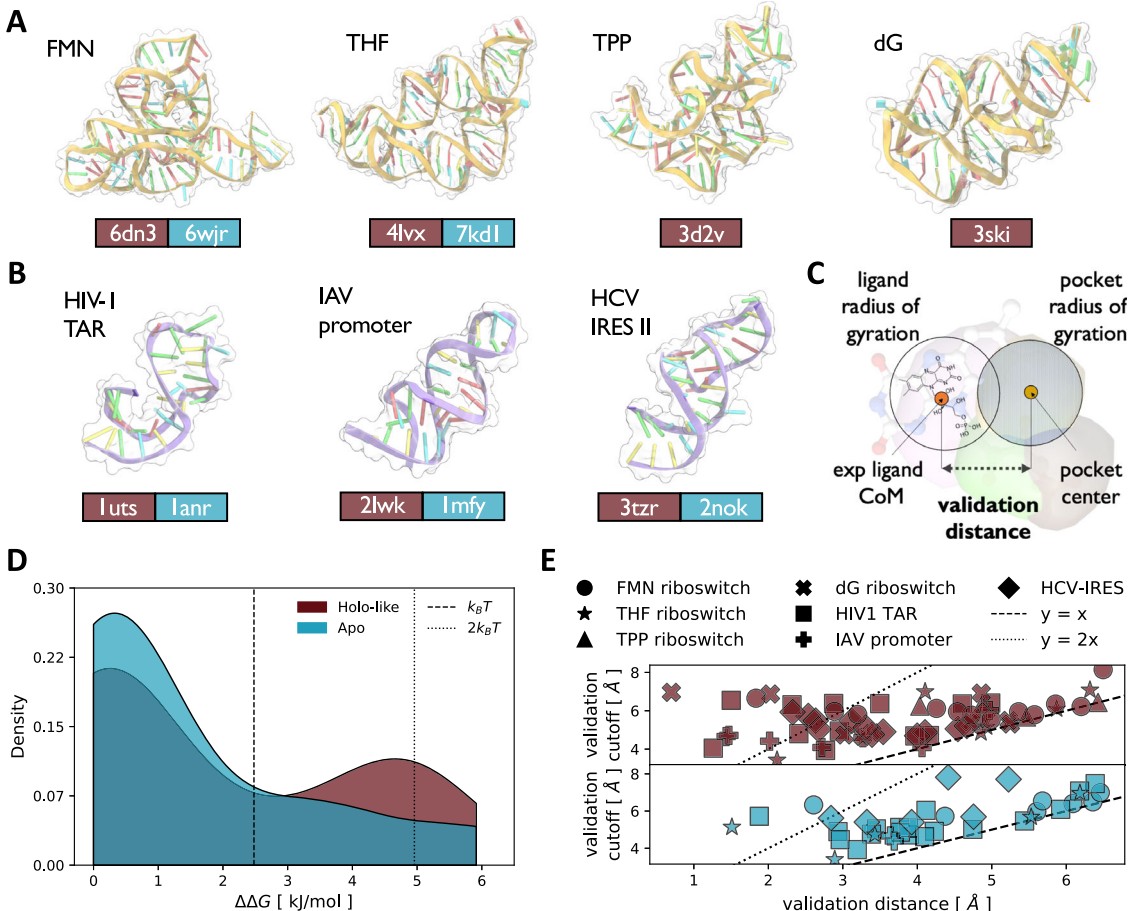

**Fig. 2 | Assessment of the SHAMAN accuracy. A** A cartoon-surface representation of the four riboswitches in our benchmark set (Tab. S1), with the corresponding name in the upper left of each panel. In the lower part, the PDB id of the starting structure used in our SHAMAN simulations is reported in a brown and cyan box for the holo-like and apo case (when available), respectively. The cartoon representations correspond to the holo-like structures. **B** As in panel (**A**), for the three viral RNAs of our benchmark set (Tab. S1). **C** Definition of the validation distance (Eq. 10) as the distance between the free-energy weighted center of an interacting site and the center of mass of the experimental ligand. **D** $\Delta\Delta G$ distribution of the probes that correctly identified known experimental pockets for holo-like (brown) and apo simulations (cyan). **E** Scatter plots of the validation distance (*x*-axis) and cutoff defined by Eq. 10 (*y*-axis) for holo-like (brown, upper panel) and apo (cyan, lower panel) simulations. The dashed line indicates validation distances equal to the validation cutoff, while the dotted line corresponds to half the validation cutoff. Each system is identified by a different marker shape, as defined in the legend.

corresponding Murcko scaffold (Methods). Our analysis did not reveal a strong correlation between ligands and probes (Tab. S9). We then calculated the Tanimoto similarity using different fingerprints (Methods). Our analysis suggested that we cannot predict whether a probe would be successful based on its similarity with a ligand (Fig. S2). However, based on a statistical classification (Methods), we can conclude that probes that did not resemble the ligand were highly unlikely to successfully identify the corresponding binding site, with a negative predictive value (NPV) equal to 0.82 (Eq. 11 and Tab. S10).

**Comparison with other tools**

We compared SHAMAN with three state-of-the-art computational tools for small-molecule binding site prediction on RNA molecules: SiteMap[49], BiteNet[50], and RBinds[51,52]. For all the systems in our benchmark set, we tested the ability of these tools to correctly predict the RNA nucleotides interacting with small molecules in experimentally determined structures (Methods). First, we determined the quality of the predictions obtained from holo-like conformations using only the corresponding experimental holo structure as ground truth (Tab. S1, red column). SHAMAN and BiteNet outperformed SiteMap and RBinds (Fig. 4A) in terms of Matthews Correlation Coefficient (MCC score), a comprehensive measure of predictive quality for binary classifiers (Methods). The low MCC scores of

SiteMap and RBinds were mostly due to their low accuracy and precision. While the quality of the predictions obtained with SHAMAN and BiteNet was comparable, the precision of our approach was more variable across our benchmark set, with a tendency to overestimate the number of interacting nucleotides. Given that SHAMAN accounts for the flexibility of the RNA target, we hypothesized that this was the result of the prediction of alternative binding pockets not present in the single holo structure used as ground truth. To verify this hypothesis, we assessed the quality of predictions by considering as ground truth for each system the set of interacting nucleotides in all the experimental binding sites of our validation set (Tab. S3 and S4, Methods). With this definition, SHAMAN precision and overall MCC score improved (Fig. 4B), in support of our hypothesis. Finally, to simulate a common drug discovery scenario in which only the structure of the apo state is available, we tested the quality of the predictions obtained from apo conformations (Tab. S1, cyan column). In this case, the quality of SHAMAN predictions was superior to BiteNet (Fig. 4C) as our approach was able to identify with high accuracy and precision the correct set of interacting nucleotides in all the reference experimental structures. These results clearly indicate that prediction tools that do not account for the flexibility of the RNA target are not able to predict binding sites formed upon local or global structural rearrangements.

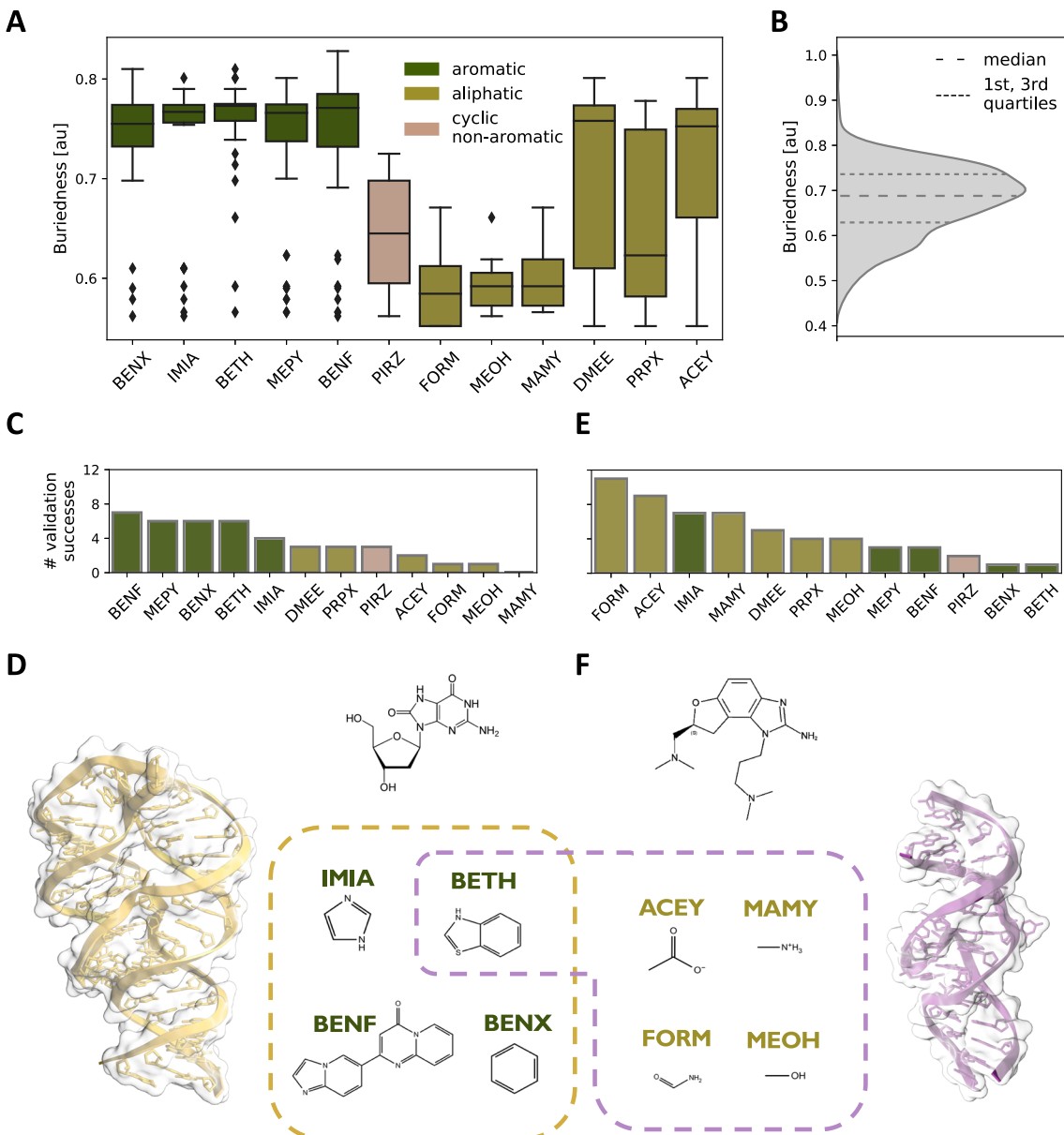

**Fig. 3 | Analysis of the SHAMAN probes. A** Violin plots representing the buriedness of the experimental pockets (*y*-axis) successfully identified by a given SHAMAN probe (*x*-axis). Buriedness values were extracted from the HARIBOSS database[22] (Tab. S3 and S4, sample size *n* = 420). Each box represents the interquartile range between the first and third quartiles, with the median indicated by a horizontal black line. Outliers are marked as black diamonds. **B** Buriedness distribution for the RNA pockets occupied by ligands in all the structures deposited in HARIBOSS. **C** Total number of times that a probe explored an experimental binding site in the riboswitches of our validation set. **D** Cartoon representation of the 2′-deoxyguanosine (dG) riboswitch (PDB 3ski) with 2D structure of the GNG binder. In the dashed box, the 2D structures of the probes that identified the GNG binding site. **E** As in (**C**), for the viral RNAs of our validation set. **F** Cartoon representation of the RNA from the Hepatitis C Virus (PDB 3tzr) with 2D structure of the SSO binder. In the dashed box, the 2D structures of the probes that identified the SSO binding site.

## The case of the FMN riboswitch

The Flavin MonoNucleotide (FMN) riboswitch is an RNA molecule found in bacteria that regulates FMN gene expression via binding the FMN metabolite[16,53]. As of today, 19 X-ray structures of the FMN riboswitch are deposited in the PDB database, 3 in apo and 16 in holo conformations. The 9 unique small molecules resolved in the holo structures fall into three main families: the cognate FMN family, the synthetic ribocil family, and the tetracyclic DKM binder (Fig. S3). The ligands belonging to the FMN and ribocil families share a U-shaped conformation and occupy the same binding site, buried into the RNA structure within the junctional region of the six stems between the A-48 and A-85 bases (Fig. 5A). The DKM tetracyclic ligand exhibits instead a distinct binding mode[54] as it induces a flip in A-48 and stacks

face-to-face between A-48 and G-62, resembling the apo form (Fig. 5B). We therefore challenged our SHAMAN approach to capture the local rearrangements of the FMN riboswitch and to identify both types of binding poses starting from a single static structure.

We tested SHAMAN starting from both holo-like (PDB 6dn3[55]) and apo (PDB 6wjr[53]) structures (Fig. S5CD). One major RNA cluster, including the initial conformations, was populated for 99% and 84% of the holo-like and apo trajectories. This limited conformational variability observed in our simulations is consistent with the structural variety resolved experimentally (Tab. S11), supporting the accuracy of the force field used in our SHAMAN simulations. In this predominant RNA structural cluster, our method successfully located the experimental binding sites (Fig. S5CD) with very high accuracy, in the best

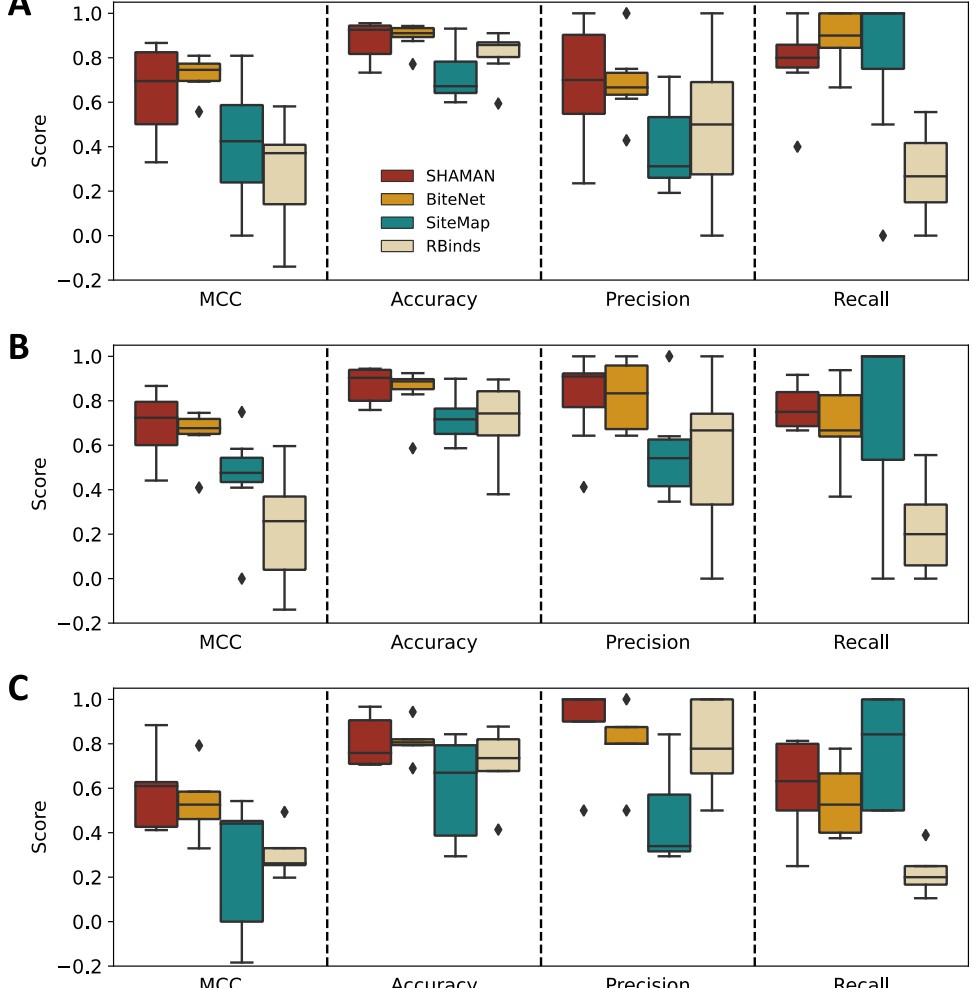

**Fig. 4 | Comparison with other tools.** From left to right, boxplots reporting the predictive quality of different binding site prediction tools evaluated by four statistical metrics for binary classifiers (Methods). **A** Binding site prediction on the holo-like systems (Tab. S1, red column, sample size $n = 7$) validated against the single corresponding experimental structure. **B**, **C** Binding site prediction on holo-like (**B**) and apo (**C**) systems (Tab. S1, red and cyan columns) against all the validation structures (Tab. S3 and S4, Methods, sample size $n = 69$). Each box represents the interquartile range between the first and third quartiles, with the median indicated by a horizontal black line. Outliers are marked as black diamonds.

case with a discrepancy of only 1.5 Å and 1.7 Å in the holo-like and apo simulations, respectively (Tab. S5). Moreover, the experimental pocket was ranked in both cases among the most probable SHAMAPs (Fig. 2D), with a ΔΔG (Eq. 9) of 0.04 kJ/mol and 0.08 kJ/mol, respectively (Tab. S5). These results are even more remarkable if we consider the buried character of the FMN riboswitch pocket, which made it difficult for the probes to access it and sample accurately. As discussed above (Fig. 3), most of the probes that successfully identified this buried pocket were aromatic, both in the holo-like (83%) and apo (75%) cases (Fig. 5E).

Notably, the two distinct binding modes of FMN and DKM ligands were identified with comparable accuracy in both runs starting from holo-like and apo conformations. Each of these starting conformations was representative of one single binding mode: in the holo-like structure, the A-48 basis faces A-85, while in the apo case it is flipped onto A-49. SHAMAN enabled the identification of both binding modes, including the one not present in the starting conformation, something not possible with algorithms based on static structures. This is highlighted by superimposing the SHAMAPs found in the holo-like and apo simulations to the corresponding starting structure (Fig. 5CD, insets). The detection of both binding modes was made possible by simulating different probes in parallel and allowing for induce-fit effects in the

RNA conformation sampled by the mother simulation (Discussion). In the holo-like case, the BENX and IMIA probes captured the tail of the FMN binder (left panel, Fig. 5F, black and green surfaces, respectively), while BENF and MEPY overlapped with the tetracyclic part of DKM (right panel Fig. 5F, orange and celeste surfaces, respectively). In the apo case, MEPY interacting site overlapped with both ligands, but the tetracyclic part of DKM was captured only by IMIA (Fig. 5G).

**The case of HIV-1 TAR element**
The HIV-1 Trans-activation response element (HIV-1 TAR) is a highly flexible, non-coding RNA molecule responsible for regulating HIV-1 gene expression through binding with Tat protein[56,57]. Understanding its conformational dynamics is crucial for drug development but remains challenging due to the major structural changes occurring upon binding diverse partners[58,59]. This conformational plasticity of HIV-1 TAR is reflected in the >20 resolved structures, primarily by NMR, alone or bound to different ligands in water-exposed cavities. Our validation set was composed of 5 holo structures bound to different small molecules with different binding modes (Fig. S4) in the groove between the bulge UCU and the apical loop CUGGGA (residues 23–25 and 30–35, Fig. 6A). This is a crucial region that also encodes the Tat protein binding site[60]. One of these structures (PDB 2l8h) indicates the

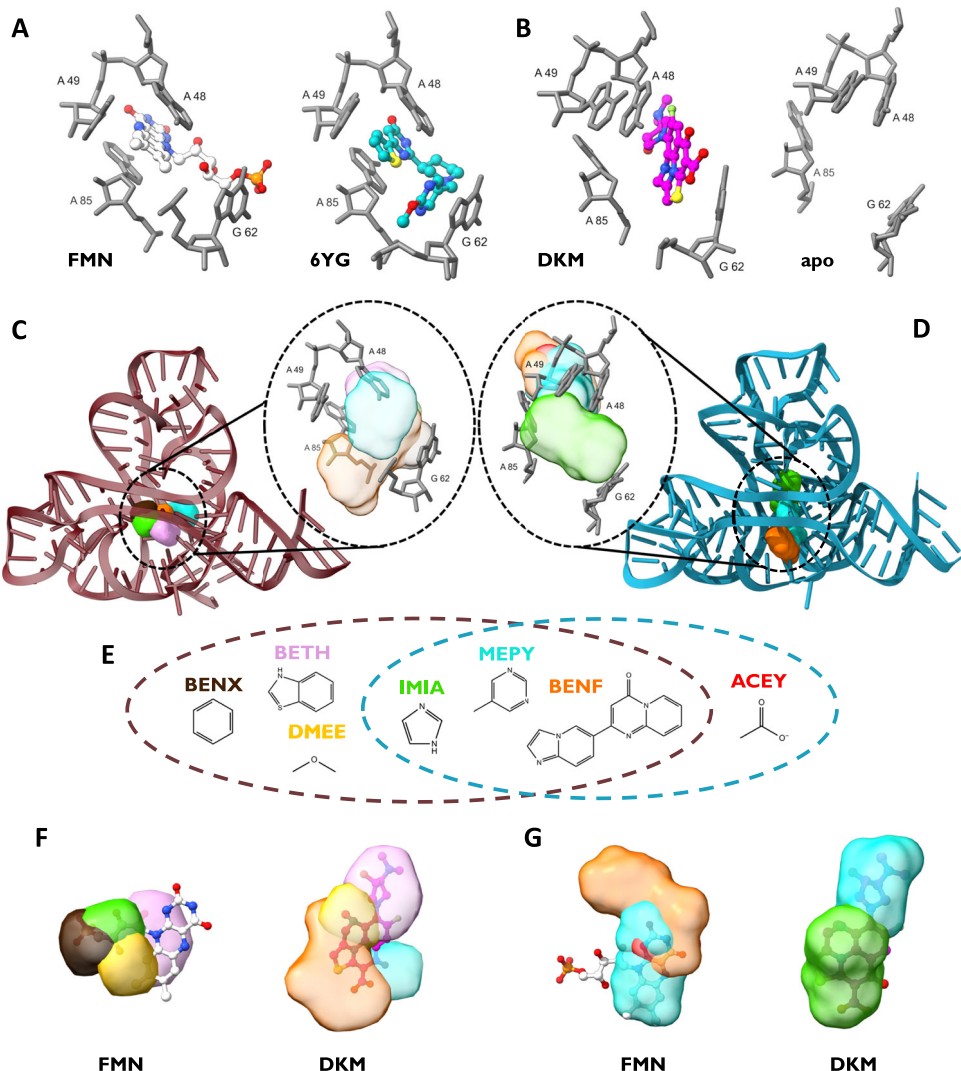

**Fig. 5 | The case of the FMN riboswitch. A** Key RNA binding site residues for the FMN ligand (PDB 2yie) and ribocil (PDB 5kx9) families. **B** Key RNA binding site residues for the DKM ligand (PDB 6bfb) and in the apo conformation (PDB 6wjr). **C, D** Cartoon representation of holo-like (**C**) and apo (**D**) starting structures used in the SHAMAN simulations of the FMN riboswitch. In the insets, the key binding site residues are overlayed with the probe densities (colors as in Tab. S7 and S8). **E** 2D

structures of the probes that successfully identified the experimental binding sites in the FMN riboswitch. The brown and cyan dashed circles indicate the successful probes in the holo-like and apo simulations, respectively. **F, G** For the holo-like (**F**) and apo (**G**) simulations, the SHAMAPs with best overlap with FMN (left) and DKM (right) ligands, representing the two different binding modes of the FNM riboswitch.

presence of a transient and functionally relevant pocket formed upon binding to the MV2003 small molecule[58]. Given its complex dynamics, HIV-1 TAR constitutes an important benchmark of the capabilities of SHAMAN to detect binding sites appearing upon global conformational changes of the target molecule.

We tested SHAMAN starting from two structures of HIV-1 TAR, one in holo-like (PDB 1uts[61]) and one in apo (PDB 1anr[62]) conformation. Both simulations recapitulated the expected flexibility of the target by identifying multiple significantly populated structural clusters (Fig. S5BC). A significant portion of the SHAMAPs was in the major groove of HIV-1 TAR (Fig. S6BC) with a relatively high probability ($\Delta\Delta G$ within $2k_BT$). Among these, SHAMAN identified all the 5 experimental binding sites, even though the overall similarity of the RNA to the deposited structures was never below ~3 Å backbone RMSD (Fig. S5). The most accurate overlaps with the experimental ligands were obtained with SHAMAPs detected in conformations $b$ and $e$ in the holo case (Fig. 6D) and conformations $a$, $c$, and $d$ (Fig. 6E) in the apo case, mostly by aliphatic probes (Fig. 6F). The geometric accuracy in identifying the binding sites was inferior compared to the FMN riboswitch, with an

average distance between binding sites equal to 4.0 Å and 4.1 Å for the holo-like ad apo cases, respectively (Table S6). However, we consider this distance still acceptable given the high flexibility of the molecule and the shallow nature of the experimental binding sites.

Notably, SHAMAN was able to identify the cryptic binding pocket proposed by Davidson et al. [58]. (orange residues in Fig. 3B of their publication). In our simulations, this site was detected in conformation $e$ (orange residues in Fig. 6C) by the ACEY and MAMY probes (red and pink densities, respectively). While in the work of Davidson et al. the cryptic pocket appeared upon MV2003 binding to HIV-1-TAR, here its detection was made possible by the ability of SHAMAN to describe large conformational changes of small RNAs and account for induce-fit effects of the probes (Discussion).

## Discussion

Here we presented SHAMAN, a computational technique for small-molecule (SM) binding site identification in RNA structural ensembles based on all-atom MD simulations accelerated by metadynamics. We benchmarked the accuracy of our approach using a set of known RNA-

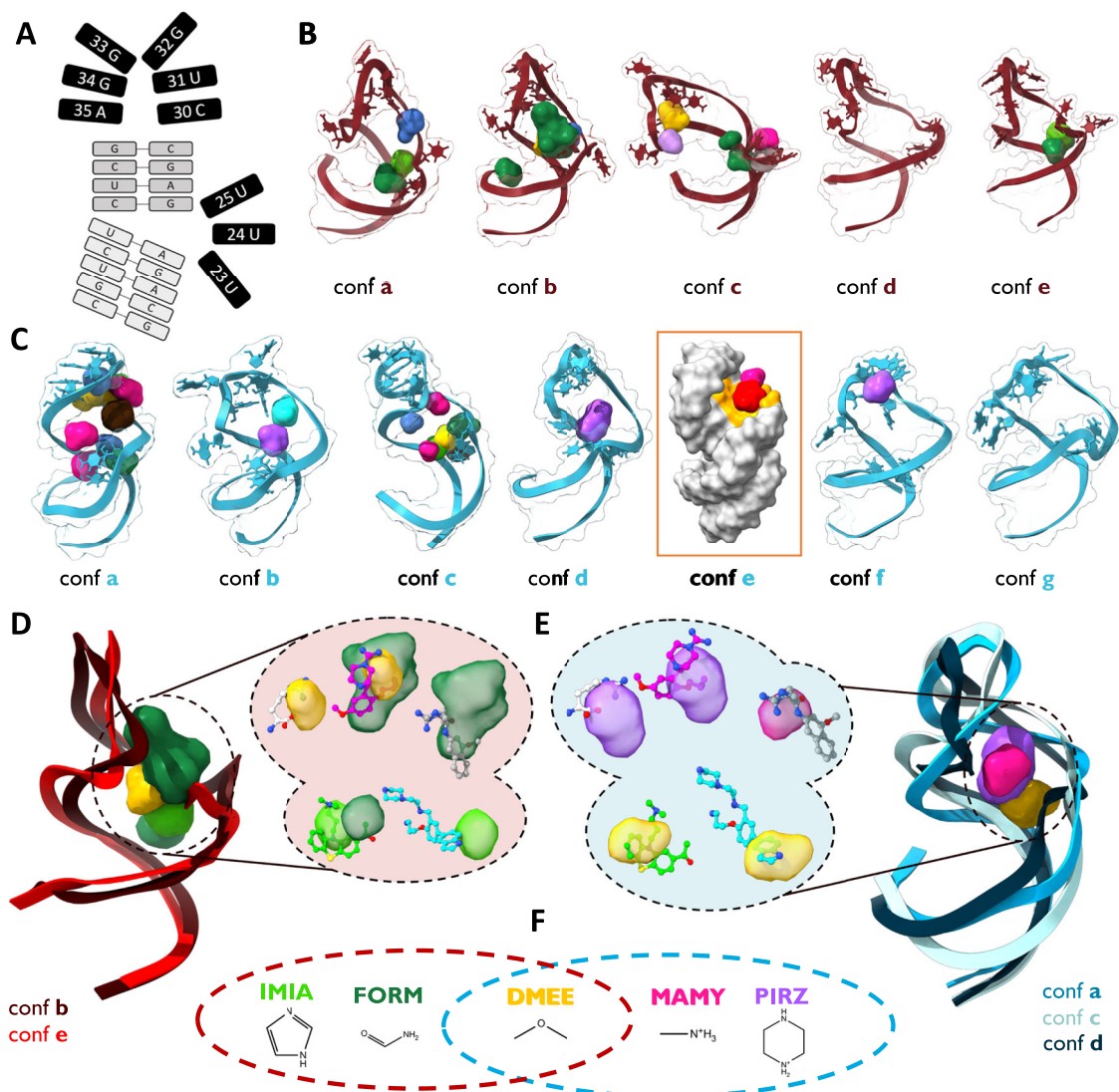

**Fig. 6 | The case of the HIV-1 TAR. A** 2D structure of the HIV-1 TAR. The two stem regions are indicated in light gray; the bulge (residues 23–25) and the apical loop (residues 30–35) in black. **B–C** Representative RNA clusters determined by the SHAMAN simulations initiated from the holo-like (**B**) and apo (**C**) conformations. SHAMAPs are visualized as solid surfaces with the color code defined in Tab. S7 and S8. The RNA state labeled as "conf e" in panel C is represented as a gray surface to highlight the orange region explored by ACEY (red density) and MAMY (rose density). This area corresponds to the cryptic binding site identified by Davidson et al. [58]. **D, E** Representative RNA conformations and SHAMAPs with best overlap with the experimental binding sites found in the simulations initiated from the holo-like (**D**) and apo (**E**) conformations. In the insets, SHAMAPs that best identified the 5 ligands present in our validation set (Tab. S4): clockwise from top left, ARG in PDB 1arj, PMZ in PDB 1lvj, P13 in PDB 1uts, P12 in PDB 1uui, MV2003 in PDB 2l8h. **F** 2D structures of the probes that successfully identified the experimental binding sites. The brown and cyan dashed circles indicate the successful probes in the holo-like and apo simulations, respectively.

small molecule structures, which included large, stable riboswitches and smaller, highly flexible viral RNAs. SHAMAN was able to identify all the binding pockets observed in the experimental structures and rank them among the most favorable probe interacting hotspots, both when starting from holo-like and apo conformations of the target. The interacting sites found by the SHAMAN simulations initiated from holo-like conformations were closer to the experimental pockets than those found in the apo cases. However, in the latter case the SHAMAPs corresponding to experimental binding sites were still very accurate and ranked as the top scored interacting sites for the majority of systems. Furthermore, our predictions were more accurate in the case of rigid riboswitches, with the regions explored by the probes perfectly matching the experimental binding sites. The accuracy was very satisfying also for viral RNA molecules considering their high flexibility.

SHAMAN emerges as one of the most advanced physics-based approaches for binding site identification in RNA structural ensembles.

A major limitation of existing CADD tools in this framework is the inadequate treatment of RNA flexibility. In these regards, SILCS-RNA[29] represents the state-of-the-art computational techniques by modeling the flexibility of the target RNA using a mixed-solvent MD approach. However, the method proposed by the MacKerell group presents two important limitations. First, it makes use of positional restraints on the RNA backbone atoms and therefore is not designed to detect cavities formed upon major conformational changes. Second, SILCS-RNA was tested only by starting the MD simulations from holo structures after the removal of the bound ligand, therefore restraining the RNA target in a conformation in which the binding site is already formed. On the contrary, SHAMAN has been designed to enable the identification of pockets in dynamic RNA conformational ensemble characterized by both local and global conformational changes. The FMN riboswitch case study highlights how the target RNA molecules simulated in the replica systems have enough freedom to undergo local rearrangements induced by the probes and ultimately to capture the two distinct

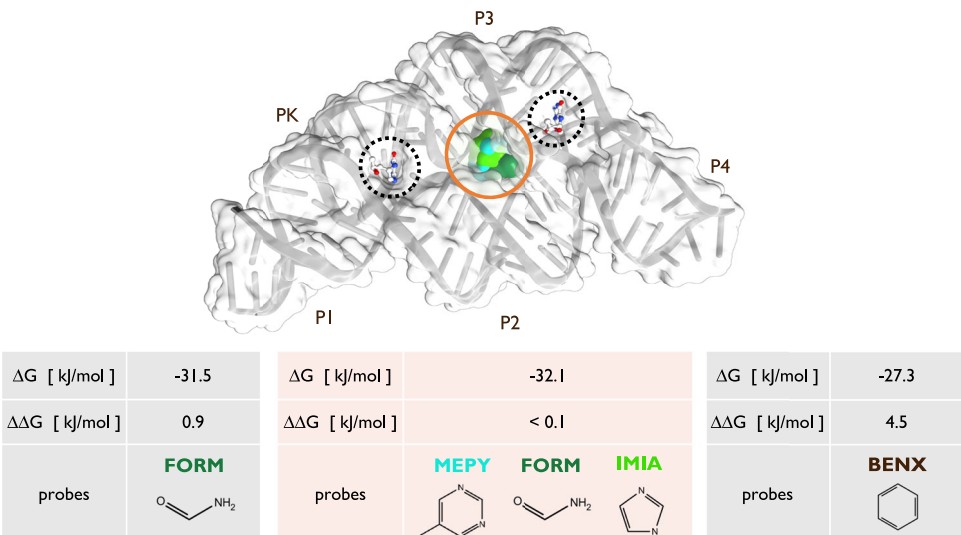

| ΔG [ kJ/mol ] | -31.5 | | ΔG [ kJ/mol ] | -32.1 | | | ΔG [ kJ/mol ] | -27.3 |
|---|---|---|---|---|---|---|---|---|
| ΔΔG [ kJ/mol ] | 0.9 | | ΔΔG [ kJ/mol ] | < 0.1 | | | ΔΔG [ kJ/mol ] | 4.5 |
| probes | **FORM** | | probes | **MEPY** | **FORM** | **IMIA** | probes | **BENX** |

**Fig. 7 | Identification of an alternative pocket in the THF riboswitch.** In the upper panel, cartoon representation and molecular surface of the center of the most populated RNA cluster found in the SHAMAN simulation initiated from a holo-like conformation (PDB 4lvx). The THF riboswitch presents two binding pockets (dashed circles), one in a three-way junction (HB4 ligand bound between helical domains P2, P3 and P4, right side) and the other in a pseudoknot (HB4 ligand bound in PK region, left side). The experimental ligands in PDB 4lvx are superimposed by aligning the coordinates to the RNA cluster center. Our protocol detected a low free-energy SHAMAP in the middle of the THF riboswitch between helix P2 and P3 (surfaces surrounded by orange circle, colored as defined in Tab. S7 and S8),. In the lower panel, the light gray and light orange tables report the details of the SHA-MAPs that identified the two experimental and the alternative binding sites, respectively.

binding modes observed in the experimental structures. Furthermore, the challenging case study of HIV-1 TAR demonstrates that cryptic pockets formed upon global conformational rearrangements[58] can also be successfully identified by SHAMAN.

Despite the potentialities discussed above, the current implementation of SHAMAN presents two important limitations. First, the unbiased MD simulation of the RNA target in the mother replica will hardly ever provide a comprehensive exploration of the conformational space at low computational cost. However, this might not be a severe limitation if the scope is to determine potential druggable sites in the proximity of the metastable holo-like and apo RNA conformations resolved experimentally. To achieve a more global conformational exploration, in the future we will accelerate sampling of the RNA target in the mother replica by using enhanced-sampling techniques distributed with the PLUMED library, where SHAMAN is also implemented. Another limitation of our approach resides in the accuracy of the RNA force fields used in our MD simulations. Despite tremendous progress[63], the accuracy of molecular mechanics force fields for nucleic acids is still as high as for proteins. One way to effectively improve the underlying force field is to integrate experimental data into MD simulations. A large variety of integrative approaches, often based on Maximum Entropy and Bayesian principles[64] have been developed in the past 10 years to use ensemble-averaged experimental data, such as many NMR observables, to model accurate structural ensembles of dynamic proteins. These approaches have been more recently applied to the determination of RNA structural ensembles[47,65] and can be used in the future to improve the accuracy of the RNA ensembles determined by SHAMAN. However, it should be noted that in the current implementation of SHAMAN the probe (pseudo) binding free energy is calculated without accounting for the population of the RNA structural cluster in which the binding site is found. Therefore, improving the cluster populations by means of integrative approaches will not have a significant impact on the accuracy of SHAMAN, provided that the sampling of the conformational landscape of RNA molecules is exhaustive in the first place.

In the future we foresee multiple different applications of SHAMAN in the context of CADD, in particular in combination with virtual screening applications and fragment-based drug design[66]. Here our approach was used only to identify binding sites occupied by ligands in experimentally resolved structures. In this process, we also detected potential alternative binding sites that were in many cases ranked among the top scored SHAMAPs. For example, in the case of the THF riboswitch, we identified a top scored SHAMAP at the center of the RNA molecule between helix P2 and P3 (Fig. 7). In this region, to our knowledge, no binders have been experimentally determined yet. In the future, we will attempt at experimentally validating this pocket and eventually targeting it in a virtual screening campaign. Even more exciting is the application of SHAMAN to novel targets for which a small molecule has not been found yet. In these regards, the fact that top scored SHAMAPs often corresponded to known binding sites will allow us to restrict virtual screening campaigns to a few localized regions.

Despite the fact that we did not find a strong correlation between successful probes and ligands, we believe that SHAMAN can provide some guidance to tailor the choice of small molecules for virtual screening or to optimize known ligands. For example, in the case of riboswitches characterized by buried cavities and viral RNA with shallower and more exposed cavities, the results of our analysis suggested the use of molecules rich in aromatic or non-aromatic moieties, respectively. In addition, areas close to the location of known ligands identified by certain probes as strong interacting hotspots could provide insights about how to modify the ligand to improve its affinity or even clues about ligand binding pathways (Fig. S6).

One of the growing concerns with rational drug discovery approaches for RNA targeting is selectivity. Although in the present study we apply SHAMAN to RNA molecules with low sequence identity, one could consider employing our protocol to examine the uniqueness of a binding site in one target against a set of undesirable targets close in sequence (antitargets). In the case where a binding site is located in the same area across all examined RNA molecules, but it has different physico-chemical and structural properties, a cross-docking approach, i.e., docking to multiple RNAs and selecting molecules with predicted affinity for the desired target significantly higher compared to the others, can be used to identify potentially selective compounds.

In conclusion, our method provides a promising foundation for future drug design efforts targeting RNA. The accuracy, reliability, and versatility of SHAMAN in identifying small-molecule binding sites across diverse RNA systems with various degree of flexibility highlight its potential value in the field. By integrating SHAMAN in virtual screening pipelines, we aim in the future at creating an advanced platform for the rational in silico design of RNA-targeting molecules, effectively addressing the longstanding challenges in the field.

## Methods

### Details of the SHAMAN algorithm

SHAMAN consists of four main stages, each one composed of a set of operations described in detail in the following sections. At the beginning of each stage, we provide a brief non-technical overview to facilitate the reading.

**Input stage.** The initial input of SHAMAN consists of the 3D structures of the target RNA and of a set of $N$ probes. Starting from this information, we generate a reference mother system, including the RNA and possibly structural ions, and $N$ replicas, each one with the addition of a different probe.

**Setup of the mother simulation.** The 3D structures of all the systems (Table S1) were obtained from the PDB database[67]. In the case of RNA structures determined by NMR, the first model was selected. In case of holo structures, the ligand was removed. Furthermore, to correctly model the RNA with our forcefield, the following elements were also eliminated, if present: crystal waters, PO3 group in the 3' terminal, modified residues at both terminals, and ions not modeled by our forcefield (SO4 in PDB 3tzr, 3ski and 7kd1). The resulting model was then prepared by adding hydrogen atoms using UCSF Chimera[68] at pH = 7.4 and processed by the OpenMM library[69] v. 7.7.0 to generate an initial configuration and topology files. The forcefield used for RNA was AMBER99SB-ILDN*[70] with the BSC0 correction on torsional angles[71] and the $\chi_{OL3}$ correction on anti-g shifts[72]. Ions were modeled using the Joung and Cheatam parameters[73] with the Villa et al. correction for magnesium[74]. Water molecules were modeled with the OPC force field[75]. Forcefield parameters were obtained from https://github.com/srnas/ff.

**Setup of the replica simulations.** The 3D structures of the probes were generated as described in the section *Details of the probes*. One replica of the system was generated for each probe. A single probe was inserted in a random position and orientation, with maximum distance of its center of mass from the RNA atoms equal to 1.0 nm. The force field and topology of the probe were created with OpenFF Sage 2.0[76].

**General details of the MD simulations.** Both mother and replica systems were solvated in a triclinic box with dimensions chosen in such a way each edge of the box was 1.0 nm away from the closest RNA atom. K+ and Cl- were added to ensure charge neutrality at salt concentration equal to 0.15 M. In all simulations the equations of motion were integrated by a leap-frog algorithm with timestep equal to 2 fs. The smooth particle mesh Ewald[77] method was used to calculate electrostatic interactions with a cutoff equal to 0.9 nm. Van der Waals interactions were gradually switched off at 0.8 nm and cut off at 0.9 nm. All simulations were performed with GROMACS[78] v. 2021.5 equipped with a development version of PLUMED[79] (GitHub master branch).

**Production stage.** After independently equilibrating mother and replica systems, the SHAMAN simulation proceeds in parallel. The RNA in the mother simulation is freely evolving and the positions of the RNA backbone atoms are communicated to the replica systems. A restraint is added to the positions of the backbone RNA atoms in the replica systems to make sure that they follow like shadows the conformation sampled by the mother. To accelerate the exploration of the RNA surface, the sampling of the probe in the replica systems is enhanced by metadynamics.

**Equilibration procedure.** All systems were independently equilibrated before the production stage. This procedure consisted of (i) energy minimization with steepest descent; (ii) a 10 ns-long equilibration in the NPT ensemble using the Berendsen barostat[80] at 1 atm; (iii) a 10 ns-long equilibration in the NVT ensemble using the Bussi-Donadio-Parrinello thermostat[81] at 300 K. During the last two steps, harmonic restraints with harmonic constant equal to 400 kJ/mol/nm² were applied to the positions of the RNA backbone as well as probe atoms.

**SHAMAN simulations.** The systems were simulated in parallel for 1 μs each. The following settings were implemented using PLUMED. First, the position of the atoms of the RNA backbone in the mother system were communicated to all the replicas with a stride equal to 0.2 ps and the corresponding atoms were restrained to have a maximum RMSD of 0.2 nm from the mother configuration using an upper harmonic wall with intensity equal to 10000 kJ/mol/nm². Second, to accelerate the probe exploration of the RNA surface, we used metadynamics[48]. As collective variables $S(R)$, we used the *xyz* coordinates of the center of mass of the probe, defined after aligning the atoms of the RNA backbone to the initial reference conformation using the FIT_TO_TEMPLATE action in PLUMED. The well-tempered variant of metadynamics[82] was used with biasfactor equal to 10. Gaussians with initial height of 1.2 kJ/mol and width of 0.1 nm were deposited every 1 ps. Finally, we restrained the position of the center of mass of the probe to be at most 1.0 nm away from the closest RNA atom using an upper harmonic wall with intensity equal to 10000 kJ/mol/nm².

**Analysis stage.** For each representative cluster of RNA conformations explored by SHAMAN, we (i) identified the regions with high probe occupancy; (ii) defined a set of potential interacting sites for each probe; (iii) clustered together the sites found by all probes to create the final SHAMAPs.

**Metadynamics reweighting.** We removed the effect of the metadynamics bias potential on the probe trajectories by calculating for each frame the unbiasing weight $w_t$ as[83]:

$$w_t \propto exp \frac{V_G(S(R_t),\bar{t})}{k_B T} \tag{1}$$

where $V_G(S(R_t),\bar{t})$ is the well-tempered metadynamics potential accumulated at the end of the simulation $\bar{t}$ and evaluated on the conformation $R_t$. All these operations were performed independently for each simulation using the *driver* utility of PLUMED.

**RNA clustering.** We first concatenated all the trajectories of the mother and replica simulations, after removal of probes, water and ions, and fixed the discontinuities due to the periodic boundary conditions. We then clustered all the RNA conformations with the *gromos* algorithm[84] implemented in GROMACS using as metrics the RMSD calculated on the RNA backbone atoms with a cutoff of 0.3 nm. To reduce memory requirements, the clustering was first performed on a subset of frames (1 every 10) and then the excluded frames were assigned to the closest cluster using a python script based on the MDAnalysis library[85] v. 2.2.0. The cluster center was taken as the representative structure for each state. The cluster populations were calculated independently for the mother and each replica simulation and clusters populated <10% were discarded in the subsequent analysis.

**Calculation of probe free energy maps.** The following analysis was performed independently for each replica and probe system as well as for each RNA cluster. We first extracted from each trajectory the frames corresponding to the selected cluster and aligned all the conformations to the RNA backbone atoms of the cluster center. We then defined a grid in the 3D space with voxel size equal to 0.1 nm and computed for each voxel $ijk$ the corresponding probe binding free energy $\delta G_{ijk}$ as:

$$\delta G_{ijk} = -k_B T \log \frac{N_{ijk}}{N_0} \tag{2}$$

where $k_B T = 2.494339$ kJ/mol and $N_{ijk}$ is the sum over all probe atoms of the (normalized) metadynamics unbiasing weights (Eq. 1) of the frames in which that atom explored the voxel $ijk$. $N_0$ is the probe occupancy in the bulk solvent:

$$N_0 = n_{probe} \frac{V_{voxel}}{V_{MD}} \tag{3}$$

where $n_{probe}$ is the number of probe atoms, $V_{voxel}$ and $V_{MD}$ the volume of the voxels and simulation box, respectively. $\delta G_{ijk}$ quantifies the propensity of finding a probe atom within the voxel $ijk$ rather than in the bulk solvent: voxels with low value of $\delta G_{ijk}$ represent therefore potential strong binding sites to the RNA molecule. We estimated the associated error $\sigma_G$ by calculating the standard deviation of $\delta G_{ijk}$ calculated in the first and second half of the trajectory (Fig. S7).

**Voxels selection, clustering into interacting sites, and filtering.** For each probe, we first selected all the voxels within 10 kJ/mol from the minimum value of $\delta G_{ijk}$ across all voxels in order to exclude weak affinity regions. The selected voxels were then clustered into *interacting sites* using the DBSCAN algorithm implemented in the scikit python library[86] v. 1.8.1, with a maximum distance between points equal to 0.2 nm and a minimum number of samples equal to 5. For each interacting site, we calculated the associated binding free energy $\Delta G_l$:

$$\Delta G_l = -k_B T \log \sum_{ijk} p_{ijk} \tag{4}$$

where $p_{ijk} = \exp[-\frac{\delta G_{ijk}}{k_B T}]$ and the sum is over all the voxels belonging to the site. For each interacting site, we also defined its center $\boldsymbol{g}_l$ as the free-energy weighted average position of the voxel centers $\boldsymbol{r}_{ijk}$:

$$\boldsymbol{g}_l = \frac{\sum_{ijk} p_{ijk} \boldsymbol{r}_{ijk}}{\sum_{ijk} p_{ijk}} \tag{5}$$

and a free-energy-weighted radius of gyration $R_l$ as:

$$R_l = \sqrt{\frac{\sum_{ijk} \left[ p_{ijk} \cdot d\left(\boldsymbol{r}_{ijk}, \boldsymbol{g}_l\right)^2 \right]}{\sum_{ijk} p_{ijk}}} \tag{6}$$

where $d$ is the Euclidean distance. Finally, we calculated the buriedness score $x_{bur}^l$ of an interacting site to quantify its exposure to solvent. For each voxel $ijk$, we first defined the RNA density $N_{ijk}^{RNA}$ as the sum of the metadynamics unbiasing weights (Eq. 1) of the frames in which an RNA atom explored the voxel $ijk$. We then defined $x_{bur}^l$ as:

$$x_{bur}^l = \frac{100}{N_l} \sum_{ijk} N_{ijk}^{RNA} \tag{7}$$

where the sum runs over all the $N_l$ voxels at the surface of the interacting site. Interacting sites with low buriedness score correspond to

regions surrounded by few RNA atoms, i.e. exposed to solvent. All the sites with buriedness score <0.15 were filtered out.

**Calculation of the final SHAMAPs.** For each representative cluster of RNA conformations, we defined a set of SHAMAPs by clustering together all the interacting sites found by all probes. To perform this operation, we used the DBSCAN algorithm applied to the centers of the interacting sites $\boldsymbol{g}_l$, with maximum distance between points given by $2^*\left[\bar{R}_l + \sigma_R\right]$, where $\bar{R}_l$ is the average radius of gyration across all sites and $\sigma_R$ their standard deviation, and a minimum number of samples equal to 1. For each SHAMAP, we defined the binding free energy $\Delta G_S$ as the minimum free energy over all the interacting sites that clustered into this SHAMAP:

$$\Delta G_S = \min_{l \in S}\{\Delta G_l\} \tag{8}$$

and $\Delta\Delta G_S$ has the difference between the binding free energy of a SHAMAP and the minimum value across all SHAMAPs (*top scored*):

$$\Delta\Delta G_S = \Delta G_S - \min_S\{\Delta G_S\} \tag{9}$$

**Output stage.** The SHAMAPs obtained at the end of the previous stage constitute the final set of hotspots associated to a given conformational state of the RNA target. The SHAMAPs are reported in a table and ordered by $\Delta G_S$. Along with this information, each SHAMAP is annotated with the properties of its constituent interacting sites: a list of probes that explored the region, their correspondent $\Delta G_l$, the population of the RNA cluster in which the site has been visited, the coordinates of the centers $\boldsymbol{g}_l$ and the radius of gyration $R_l$.

## Details of the SHAMAN benchmark

**Details of the target RNAs.** For our SHAMAN simulations, we selected 7 RNA systems, whose structures in complex with at least one ligand were deposited in the PDB databank[67] (Tab. S1). To initiate the simulations, we selected 1 holo structure per system and, when available, an apo structure of the same RNA molecule. In total we performed 12 SHAMAN simulations. A summary of all simulations performed along with details about the systems are reported in Tab. S2.

**Details of the PDB structures used for validation.** To benchmark the accuracy of our approach, we first retrieved for each system all the holo structures deposited in the PDB with different ligands and binding poses. We then visually inspected each structure and identified 14 structures with unique binding poses and pockets. All the structures used for validation along with details about the RNA, the ligand, and the experimental method and resolution are reported in Tab. S3 and S4.

**Details of the probes.** The set of probes used in our protocol is composed of two subsets. First, we included 8 probes already used in the SILCS-RNA study[29], namely acetate (ACEY), benzene (BENX), dimethyl-ether (DMEE), formamide (FORM), imidazole (IMIA), methyl-ammonium (MAMY), methanol (MEOH), and propane (PRPX) (Tab. S7). These fragments had been selected in the original study as a representative set of functional groups. Second, we developed the following approach to identify fragments with higher probability to bind to RNA molecules. Two databases were used, namely HARIBOSS[20] comprising 265 experimentally validated RNA binders (https://hariboss.pasteur.cloud) and RBIND[24] that includes 159 RNA bioactive molecules (https://rbind.chem.duke.edu). In an effort to identify chemical groups that exist in both libraries, we prepared the Murcko scaffolds from the molecules derived from both databases and compared the corresponding sets. 6 Murcko scaffolds appear in both HARIBOSS and RBIND molecules (Tab. S7). From these, 5 representative scaffolds were selected for the SHAMAN simulations, namely

benzene (BENX), dihydro-pyrido-pyrimidinone-imidazo-pyridine (BENF), benzothiophene (BETH), methyl-pyrimidine (MEPY), and piperazine (PIRZ). The preparation and comparison of the HARIBOSS and RBIND libraries was done using a KNIME 4.6 protocol that includes the following steps: (i) molecule preparation using Epik[87] at pH 7.4, (ii) conversion to canonical SMILES using RDkit v. 2022.3, (iii) Murcko scaffold derivation using the RDkit Murcko Scaffolds KNIME node, (iv) set comparison using the 'Compare Ligand Sets' node provided by Schrodinger v. 2022.3, and finally (v) a fragmentation of the common scaffolds using the RECAP fragmentation method[88] (implemented as the 'Fragments from Molecules' node provided by Schrodinger). All probes used in the SHAMAN simulations have been prepared using the LigPrep module of Schrodinger Suite[89] at pH 7.4. BETH was intentionally modeled in a protonated state, as it appears in the origin molecules from RBind and HARIBOSS.

**Details of the validation procedure.** To benchmark the accuracy of our approach in identifying binding sites occupied by a ligand in known experimental structures, we used the following procedure:

i. **Multiple sequence alignment**
   For each simulated system, we aligned the sequence of our target RNA with the sequences of all the validation PDBs using CLUSTALW[90] v. 2.0.

ii. **Structural alignment of validation PDBs to SHAMAN cluster centers**
   For each validation PDB, we defined the binding site as the set of nucleotides with at least one atom within 0.6 nm of a ligand atom. The backbone atoms of the validation PDB belonging to this region were then structurally aligned to the corresponding nucleotides in each RNA cluster center, based on the sequence alignment defined above.

iii. **Definition of success for a probe interacting site**
   For each validation PDB, we defined an experimental sphere centered on the center of mass of the heavy atoms of the ligand $\boldsymbol{g}_{exp}$ and with a radius given by its radius of gyration $R_{exp}$. For each probe interacting site, we defined a validation sphere centered on the free-energy weighted center of the interacting site $\boldsymbol{g}_l$ and with radius given by its free-energy weighted radius of gyration $R_l$. We then considered a probe interacting site as successful if the validation sphere was overlapping with the experimental sphere:

$$d\left(\boldsymbol{g}_l, \boldsymbol{g}_{exp}\right) \le R_l + R_{exp} \qquad (10)$$

   In case of match with multiple validation structures, we retained only the one corresponding to the interacting site with lower $\Delta\Delta G$ from the top scored SHAMAP.

iv. **Definition of success for a SHAMAP**
   A SHAMAP was considered successful in identifying a known ligand binding site if at least one of the probe interacting sites that compose the SHAMAP was successful according to the criterion defined above (Supplementary Data 1).

**Probes-ligands comparison**
For probes and ligands in the SHAMAN simulations initiated from holo structures, we first calculated the following set of descriptors with RDKit v. 2022.3: molecular weight, number of aromatic rings, number of H-bond donors/acceptors, topological polar surface area (TPSA), and number of heterocycles. The correlation between probes and ligands descriptors was then computed with scipy v. 1.8.1 using the Pearson correlation coefficient. The analysis was performed using either the entire ligand or its Murcko scaffold. We also quantified the similarity between ligands and successful probes using different types of fingerprints (FPs) implemented in RDKit. In particular, we used Morgan (radius = 2, 2048 bits), RDKit (2048 bits), and MACCS FPs.

Using these FPs and the Tanimoto distance, we calculated the similarity between successful probes and reference ligands, considered either as entire ligands or using their corresponding Murcko scaffold.

To further investigate a possible correlation between ligand and successful probes, we formulated the following hypothesis: the ability of a probe to identify a binding site is related to its similarity to the corresponding ligand. We then compared each of the 13 probes (Tab. S7 and S8) with all the 8 ligands resolved in the experimental pockets (Tab. S1) and considered a probe to be similar (dissimilar) to a ligand if the Tanimoto distance calculated with MACCS FP was greater (lower) than 0.4 (0.2). Based on the SHAMAN results in our benchmark, we built a confusion matrix of the four possible outcomes (Tab. S10) and defined the SHAMAN negative predictive value $NPV$ as the ratio between true negatives TN and total number of negatives TN + FN:

$$NPV = \frac{TN}{TN + FN} \qquad (11)$$

**Comparison with other tools**
We selected three state-of-the-art tools for RNA binding site detection: SiteMap[49], BiteNet[50], and RBinds[52]. We evaluated the ability of these tools to predict the RNA nucleotides that belong to an experimentally detected binding site in the 7 systems of our benchmark set, including holo-like and apo structures, for a total of 12 conformations (Tab. S1).

**Definition of the ground truth.** For each system, the reference set of binding site nucleotides was defined as follows:

i. We performed a multiple sequence alignment of all the systems in our validation set (Tab. S3 and S4) using CLUSTALW[90] v. 2.0;

ii. We discarded all the nucleotides that were not resolved in all the validating structures;

iii. In each validating structure, we defined as interacting with the small molecule all the nucleotides with at least one atom within 4 Å of an atom of the ligand;

iv. To compare the predictions against all the validating structures (Fig. 4BC), we defined as interacting nucleotides the union of all the interacting nucleotides across all the validating structures.

**Prediction of interacting nucleotides.** For each software, the input was the same PDB file that was used as starting structure for our SHAMAN simulations (Details of the SHAMAN algorithm, II. Production stage). The set of predicted interacting nucleotides was defined as follows:

– **SHAMAN.** Each interacting site predicted by SHAMAN is stored in a file as the set of coordinates of the centers of the grid voxels (Details of the SHAMAN algorithm, III. Analysis stage). We defined as interacting all the nucleotides found in the RNA cluster center with at least one atom closer than 4 Å from the coordinates of all the interacting sites belonging to the SHAMAPs that identified the experimental pockets considered for validation (Tab. S5 and S6).

– **SiteMap.** For each structure, a local installation of SiteMap (v. 2023-4) was run from the command line with the options: -keep-volpts and -modbalance yes. The output was a PDB-like file containing the coordinates of the predicted binding sites. Among the predicted binding sites, we visually selected the one that was best overlapping with the position of the experimentally resolved ligand. Finally, we defined as interacting all the nucleotides with at least one atom within 4 Å of the pseudo-atoms defined in the output PDB file.

– **BiteNet.** For each structure, BiteNet was executed using a standalone version of the software. The input parameter "input probability score threshold" was set at its default value of 0.1 and the "RNA-small molecule binding site" option was selected. The binary classification of interacting/non-interacting nucleotides

was defined in the output file *"predictions.csv"*.
- **RBinds**. For each structure, RBinds was executed via the webserver available at http://zhaoserver.com.cn/RBinds/RBinds.html. The list of predicted interacting nucleotides was defined in the "sites" card in the output file *"RNAcentrality.json"*.

**Comparison metrics.** The quality of the prediction of interacting nucleotides was defined based on the following metrics for binary classifiers:

- the Matthew Correlation Coefficient (MCC), which is a global measure of prediction quality recognized for its comprehensiveness and reliability compared to other standard metrics[91]. The MCC score accounts for the quality in all the four classes of the confusion matrix:

$$MCC = \frac{TP * TN - FP * FN}{\sqrt{(TP+FP)(TP+FN)(TN+FP)(TN+FN)}} \quad (12)$$

- the accuracy, which is the fraction of correct (positive and negative) predictions:

$$accuracy = \frac{TP + TN}{TP + TN + FP + FN} \quad (13)$$

- the precision, which is the fraction of relevant instances among the retrieved instances:

$$precision = \frac{TP}{TP + FP} \quad (14)$$

- the recall (or sensitivity), which is the fraction of relevant instances that were retrieved:

$$recall = \frac{TP}{TP + FN} \quad (15)$$

**Reporting summary**
Further information on research design is available in the Nature Portfolio Reporting Summary linked to this article.

## Data availability
The GROMACS topology files and PLUMED input files used in our benchmark are available on PLUMED-NEST, the public repository of the PLUMED consortium[92], as plumID:23.031 [https://www.plumed-nest.org/eggs/23/031].

## Code availability
SHAMAN simulations can be run with the development version (GitHub master branch) of PLUMED. Scripts to facilitate the preparation of the input files and the analysis of the results as well as a complete tutorial are expected to be released soon under a license "free for academics, not for commercial use".

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

## Acknowledgements

The authors would like to thank Giovanni Bussi for advice on running MD simulations of RNA molecules; Matteo Masetti and Mattia Bernetti for providing feedback on the manuscript; Petr Popov for assistance in using BiteNet. F.P.P. was funded by Sanofi and the Association Nationale de la Recherche et de la Technologie (ANRT) contract 2020/1259. This work was granted access to the HPC resources of IDRIS under the allocation 2022-AD01101371 made by GENCI.

## Author contributions

M.B. and P.G. conceived and designed the research project. F.P.P. implemented SHAMAN, performed simulations, and analyzed the data. M.B., P.G., and F.P.P. wrote the paper.

## Competing interests

F.P. Panei and P. Gkeka are or were Sanofi employees and may own stocks in Sanofi. M. Bonomi declares no competing interests.
