## [Peer Review File · Nature Communications]

Identifying small-molecules binding sites in RNA conformational ensembles with SHAMANREVIEWER COMMENTS

Reviewer #1 (Remarks to the Author):

Non-coding RNAs are emerging drug targets, and new methods are needed to address their flexible 3D structural landscapes. The current manuscript reports a computational method called SHAMA to examine the landscape of RNA using MD simulations and for identifying binding pockets. Such computational tools are needed to advance this field. However, the current manuscript is not suitable for publication in Nat Comm for the following reasons:

1. The new method only identifies binding pockets. Still, it does not address the more difficult and important problem of finding small molecules that can bind a given pocket with high affinity and specificity. In this context, the tool presented is incremental and more appropriate for a more specialized computational journal.
2. The method relies on MD simulations yet as the authors point out, the force fields remain underdeveloped for RNA – and there are more compelling methods to determine RNA landscapes that use some input experimental data. Thus, the computational ensembles used in this approach purely derived from computation are of poorer quality than those obtained with the aid of experimental data.
3. There are other approaches for identifying pockets – given an experimentally determined landscape of RNA structures – and the authors make no critical comparisons of their approach relative to other software commonly used in RNA and protein targeting.
4. The authors select FMN and TAR as two examples – but there should be a more comprehensive analysis of RNAs, including head-to-head comparisons relative to other methods for identifying binding pockets.

I think the work is important – but the claim by the authors that “Our work constitutes to date the most advanced computational pipeline for binding site identification in dynamic RNA structural ensembles, thus providing crucial information for structure-based rational design of novel compounds targeting RNA” is an overstatement not supported by the data – in my view, even if data backed the claim, the advance does not warrant publication in a general readership journal such as Nat Comm.

Reviewer #2 (Remarks to the Author):

The study by Panei et al. introduces SHAMAN, a novel computational method aimed at addressing the challenges associated with identifying small-molecule binding sites within RNA structures. In contrast to existing tools that rely on static structures, SHAMAN utilizes atomistic molecular dynamics to explore the dynamic conformational landscape of RNA while concurrently identifying potential binding pockets through parallel probes and enhanced sampling techniques, enabling the on-the-fly determination of binding affinities. The method underwent successful benchmarking against a variety of RNA structures, encompassing both rigid and flexible RNAs, with remarkably positive outcomes. SHAMAN consistently and accurately identified experimentally confirmed binding sites, ranking them among the top candidates in all cases. The paper is a significant contribution to the field, I recommend its publication after the revision of the following comments.

While the method's explanation is thorough, it lacks the provision of tools or algorithms for readers to apply to other systems. To address this, it is recommended that the authors share a test case, and the code including input and expected output, with the scientific community upon acceptance of the paper.

One of the paper's key strengths lies in its ability to study flexible RNA systems and the importance it places on integrating RNA dynamics with ligand sampling for binding site identification. However, these claims lack empirical support. Is the treatment of RNA flexibility important? To address that I suggest that the authors rigidify at least one flexible RNA system, and repeat the study and compare the

binding behavior with the flexible counterpart and experimental data. This way, we can assess whether flexibility indeed enhances prediction accuracy or not.

The paper highlights differences in results obtained from simulations initiated with apo and holo conformations, which raises concerns regarding sampling completeness. To address this issue, it is proposed that the authors perform extended simulations for a single system to determine whether the two starting conformations converge to the same $\Delta\Delta G$ value. This analysis would help clarify the necessary simulation time for achieving thermodynamic convergence for cases where only holo-like structure is at hand.

Additionally, readers benefit from a systematic analysis on the crucial parameters influencing calculation accuracy, such as simulation time, number of probe replicas, number of clusters, metadynamics bias, etc. The authors are encouraged to select a test system and explore these parameters to determine their impact on prediction accuracy. Including these results in the supplementary information would help the readers to design their setups accurately.

Finally, while the method's success is commendable, a comparative analysis with existing methods is essential to assess its performance comprehensively. The authors need to conduct benchmarking against other methods and present a comparison to help readers make informed decisions about adopting this technique.

Reviewer #3 (Remarks to the Author):

The paper by Panei et al. reports on an innovative approach to RNA-based computational drug design, representing a noteworthy contribution to the burgeoning field of RNA therapeutics. Employing enhanced sampling simulations along with a fragment-driven exploration of RNA pockets, the authors present a promising methodology. The new approach was then applied to riboswitches and flexible viral RNAs, addressing a critical gap in RNA-based drug discovery: identifying druggable pockets for subsequent docking and free energy simulations. Given its potential significance, the paper should be considered for publication in Nature Communications. However, several substantial and minor points warrant attention and resolution before final acceptance.

1. Comparison with Existing Tools: The authors should comprehensively compare their approach and established geometrical and structural pocket-finding tools. Combining their method with existing tools may yield even richer insights and more robust results. While fragments can help in druggability, the present methodology is very sensitive to fragment chemistry.

2. Fragment Selection: The fragment selection process appears somewhat arbitrary in the current study. To significantly impact the drug discovery community, the authors should consider a more rational and chemically informed approach to fragment selection. Would you be able to comment on this point and provide a more solid justification? A table detailing the selected fragment structures would also enhance the reader's understanding. Fragment selection deserves much more detail and chemical thinking.

3. Selectivity and Multitarget Drug Discovery: Given the growing concern in RNA-based drug discovery regarding selectivity, the paper should discuss whether the new methodology offers any insights or solutions. Additionally, exploring the potential for multitarget drug discovery in the realm of RNA therapeutics would be valuable.

4. Combination with Virtual Screening: Although the text mentions combining the approach with virtual screening protocols, the specifics of this integration still need to be clarified. Clarifying how these two methods can be effectively combined would enhance the methodology's practical

applicability. One may expect the virtual screening outcomes to be very much biased by the chemical structure of the original fragments.

5. Fragment Library Screening: As an extension of point 4, the authors might consider the feasibility of screening a small library of fragments and ranking them using their new method. This could provide a practical demonstration of its utility and the general applicability of the technique. In brief, if fragments chemically out of those used for pocket detection are systematically discarded, the methodology may be weak in general applicability.

6. Free Energy Estimations: The paper needs more information on the errors associated with the reported free energy estimations. Precision in free energy values, including their associated uncertainties, should be carefully considered (how many digits)?

7. Experimental Validation: In the future, the authors should prioritize experimental validation of their predictions through techniques such as surface plasmon resonance (SPR) or microscale thermophoresis (MST), ITC and so on. Such validation is crucial for establishing the credibility and practical utility of the new methodology.

We are excited to hear the authors' response to these concerns and hope to see the further development and validation of their promising approach in the near future.

Reviewer #1 (Remarks to the Author):

Non-coding RNAs are emerging drug targets, and new methods are needed to address their flexible 3D structural landscapes. The current manuscript reports a computational method called SHAMA to examine the landscape of RNA using MD simulations and for identifying binding pockets. Such computational tools are needed to advance this field. However, the current manuscript is not suitable for publication in Nat Comm for the following reasons:

1. The new method only identifies binding pockets. Still, it does not address the more difficult and important problem of finding small molecules that can bind a given pocket with high affinity and specificity. In this context, the tool presented is incremental and more appropriate for a more specialized computational journal.

We thank the reviewer for this observation. We do agree that finding small molecules that bind a given pocket with high affinity and specificity is the ultimate goal of rational drug design. However, this goal cannot be effectively achieved with computational approaches without first exploring the conformational landscape of RNA molecules and identifying pockets with the potential to bind small molecules (regardless of their affinity and specificity). For these reasons, the step of binding site prediction in RNA structural ensembles cannot be dismissed as a minor step: it is in fact the first and fundamental step.

In direct RNA targeting with small molecules, which is the area of interest of our study, a molecule should ideally bind selectively to one RNA target and have a specific desired effect. To the best of our knowledge, the specificity of a molecule can be assessed only via experimental assays, e.g. RNA profiling or NGS. Regarding the other important issue of selectivity, we recognize that our current implementation of SHAMAN does not address this point. In our mind, the problem of selectivity can instead be addressed during the phase of virtual screening (VS) of the pockets identified by SHAMAN. During the stage of VS, small molecules can be docked *in silico* against different RNAs (i.e. cross docking) and top-ranked binders predicted to be potentially selective for the desired RNA can thus be identified. We have added to the revised version of the manuscript (Discussion) the following paragraph to comment on this issue:

“One of the growing concerns with rational drug discovery approaches for RNA targeting is selectivity. Although in the present study we apply SHAMAN to RNA molecules with low sequence identity, one could consider employing our protocol to examine the uniqueness of a binding site in one target against a set of undesirable targets close in sequence (antitargets). In the case where a binding site is located in the same area across all examined RNA molecules, but it has different physico-chemical and structural properties, a cross-docking approach, i.e. docking to multiple RNAs and selecting molecules with predicted affinity for the desired target significantly higher compared to the others, can be used to identify potentially selective compounds.”

Regarding the identification of ligands with high affinity, we agree with the reviewer that this is a crucial step in RNA drug discovery. However, we believe that optimizing ligand affinity is beyond the scope of this study. Nevertheless, our SHAMAN protocol can in principle provide useful

information to improve affinity. Often RNA targeting campaigns are based on phenotypic screenings that provide hits with low to medium affinities. In such cases, SHAMAN can be used to identify the binding sites of the hits found experimentally and thus to provide the structural information needed for docking and to perform more accurate binding affinity calculations for lead optimization, for example with Free Energy Perturbation (FEP). We are in the process of using SHAMAN exactly with this modality on a target not included in this study, the non-coding RNA MALAT1.

Finally, we would like to comment about the statement “the tool presented is incremental”, which we perceived as made with a negative connotation. Our view of science is that all research is to some extent incremental, as by nature all scientific work builds on the knowledge previously acquired by the community. There is nothing wrong with this “incremental” nature of research and there is certainly no relation between research being “incremental” and having impact on the community. One recent outstanding example is AlphaFold2. Technically, this work can be regarded as “incremental”, as there are no new technical breakthroughs in the machine learning methods implemented therein. There is instead an extreme fine-tuning of each and every component of the algorithm along with an exploitation of the previous work by the bioinformatics community (for example, the extraction of spatial information from the analysis of sequence co-evolution) as well as of the structural knowledge accumulated over the years in the PDB database. Despite the incremental nature of AlphaFold2 (from a purely technical standpoint), the results obtained with this approach and the overall impact in the field have been quite revolutionary. Similarly, and certainly on a smaller scale compared to AlphaFold2, we believe that our work, while building on techniques widespread in the computational biology community, addresses the major limitation in current approaches to identify small molecules binding sites in RNAs, i.e. the inherent flexibility of these molecules.

2. The method relies on MD simulations yet as the authors point out, the force fields remain underdeveloped for RNA – and there are more compelling methods to determine RNA landscapes that use some input experimental data. Thus, the computational ensembles used in this approach purely derived from computation are of poorer quality than those obtained with the aid of experimental data.

We thank the reviewer again for this important observation. We are well aware that integrative methods that incorporate experimental data are currently among the most accurate approaches to determine structural ensembles of biological systems, including proteins and RNA molecules. As a matter of fact, the PI of our group has been pioneering these methods since 2014 (10.1016/j.str.2014.04.019). As an example, the integrative metainference approach (10.1126/sciadv.1501177) developed by Dr. Bonomi is one of the state-of-the-art techniques to generate structural ensembles guided by experimental data and accurate atomistic force fields. These integrative methods, which we extensively reviewed in 10.1016/j.sbi.2016.12.004, allow to obtain structural ensembles in which the population of each state is more accurate compared to standard MD simulations.

However, in the current implementation of our SHAMAN approach, accurate determination of state population is not crucial. As a matter of fact, in SHAMAN the probe binding free energy that we use to rank pockets is calculated without accounting for the population of the specific RNA state in which the pocket is found, above a minimum threshold set by the user. The reason behind this choice is that, since we are simulating the RNA molecule in presence of probes, we cannot anticipate how a small molecule will modulate the population of a given RNA state. Therefore, we decided not to account for the RNA state populations when computing the probes (pseudo) binding free energies. This is a quite standard procedure when doing *in silico* ensemble docking on protein targets. In this context, the conformations of the target that are used to dock small molecules are often selected based solely on the druggability score of the identified pockets rather than the exact population of the protein states. Similarly, the pockets identified by SHAMAN are scored based solely on the pseudo-affinity of the probe for a specific conformation, regardless of its population. Since our “standard” MD simulations of the RNA molecules captured the geometry of the experimentally resolved binding pockets, even when initiated from apo conformations, any improvement in the population of the RNA clusters obtained upon integration of experimental data will not dramatically affect the overall results. Therefore, we believe that an accurate determination of the population of the RNA structural states is not a determining factor for the success of SHAMAN and consequently the use of advanced integrative approaches is not strictly necessary. On top of that, SHAMAN users can choose the force fields or advanced integrative approaches that they believe to be the most appropriate for the system of interest.

We understand that the point discussed above is very important and should be more explicitly discussed in our manuscript. We have therefore added the following sentence to the discussion section to clarify the fact that in this first implementation of SHAMAN the population of the RNA clusters are not factored in when calculating the probe (pseudo) binding free energy:

“However, it should be noted that in the current implementation of SHAMAN the probe (pseudo) binding free energy is calculated without accounting for the population of the RNA structural cluster in which the binding site is found. Therefore, improving the cluster populations by means of integrative approaches will not have a significant impact on the accuracy of SHAMAN, provided that the sampling of the conformational landscape of RNA molecules is exhaustive in the first place.”

3. There are other approaches for identifying pockets – given an experimentally determined landscape of RNA structures – and the authors make no critical comparisons of their approach relative to other software commonly used in RNA and protein targeting.

We thank the reviewer for raising this fundamental point. The reviewer is absolutely right, and we apologize for this major oversight on our side.

Despite the fact that several binding site detection tools are available for proteins, there is only a limited number of approaches that can be applied to RNA molecules. In the following, we report a list of software selected as the best performing representatives of different classes of methods, i.e. grid-based, machine learning, deep learning, and regression models.

- *SiteMap* [<https://doi.org/10.1021/ci800324m>], which is part of the Schrodinger suite, uses physics-based energy calculations to predict the most probable interaction hotspots from the structure of the target. *SiteMap* is a state-of-the-art tool for protein binding site detection and its latest release has been optimized for nucleic acids;
- *BiteNet* [<https://doi.org/10.1093/nargab/lqab111>] uses a convolutional neural network trained on known RNA-ligand structures deposited in the PDB to predict small-molecules binding sites. *BiteNet* has been shown to be more accurate in binding site identification than four other popular tools: *RSite* [10.1038/srep09179], *RSite2* [<https://doi.org/10.1038/srep19016>], *RBind* [<https://doi.org/10.1093/bioinformatics/bty345>], and *RNASite* [10.1093/bioinformatics/btaa1092]. For this reason, we decided not to explicitly test *RSite*, *RSite2*, *RBind*, and *RNASite*.
- *RBinds* [<https://doi.org/10.1016%2Fj.csbj.2020.10.043>] is built on *RBind* and constructs a graph network from the RNA structure to predict the binding regions from the closeness and degree scores between nodes. *RBinds* has also been shown to be more accurate in binding site identification than *RSite* and *RSite2*;
- *RLBind* [<https://doi.org/10.1093/bib/bbac486>] uses a deep convolutional neural network-based model to predict RNA-ligand binding sites. *RLBind* has been shown to be more accurate than *RSite*, *RSite2*, *RBind* and *RNASite*. However, this tool was not compared against *BiteNet* in the original publication. We tested the *RLBind* pipeline, which is available via a GitHub repository (<https://github.com/KailiWang1/RLBind>), and we found it not functional. Our execution was stopped at step 4.2, which was supposed to be carried out by a web server that is currently not reachable (<http://consurf.tau.ac.il>). Being the comparison impossible, the tool has been omitted from the analysis.

We compared all these methods with SHAMAN based on their ability to correctly predict the RNA nucleotides interacting with small molecules in experimentally determined structures. A detailed description of how this comparison was performed and the definition of the metrics used is reported in the revised version of our manuscript (Materials and Methods). A section named “Comparison with other tools” that illustrates the results of our analysis has been added along with the new Fig. 4. We report below the text added to the “Results” section of the main manuscript about the comparison with other tools:

“Comparison with other tools

We compared SHAMAN with three state-of-the-art computational tools for small-molecule binding site prediction on RNA molecules: SiteMap⁴⁹, BiteNet⁵⁰, and RBinds^{51,52}. For all the systems in our benchmark set, we tested the ability of these tools to correctly predict the RNA nucleotides interacting with small molecules in experimentally determined structures (Materials and Methods). First, we determined the quality of the predictions obtained from holo-like conformations using only the corresponding experimental holo structure as ground truth (Tab. S1, red column). SHAMAN and BiteNet outperformed SiteMap and RBinds (Fig. 4A) in terms of Matthews Correlation Coefficient (MCC score), a comprehensive measure of predictive quality for binary classifiers (Materials and Methods). The low MCC scores of SiteMap and RBinds were mostly

due to their low accuracy and precision. While the quality of the predictions obtained with SHAMAN and BiteNet was comparable, the precision of our approach was more variable across our benchmark set, with a tendency to overestimate the number of interacting nucleotides. Given that SHAMAN accounts for the flexibility of the RNA target, we hypothesized that this was the result of the prediction of alternative binding pockets not present in the single holo structure used as ground truth. To verify this hypothesis, we assessed the quality of predictions by considering as ground truth for each system the set of interacting nucleotides in all the experimental binding sites of our validation set (Tab. S3 and S4, Materials and Methods). With this definition, SHAMAN precision and overall MCC score improved (Fig. 4B), in support of our hypothesis. Finally, to simulate a common drug discovery scenario in which only the structure of the apo state is available, we tested the quality of the predictions obtained from apo conformations (Tab. S1, cyan column). In this case, the quality of SHAMAN predictions was superior to BiteNet (Fig. 4C) as our approach was able to identify with high accuracy and precision the correct set of interacting nucleotides in all the reference experimental structures. These results clearly indicate that prediction tools that do not account for the flexibility of the RNA target are not able to predict binding sites formed upon local or global structural rearrangements.”

4. The authors select FMN and TAR as two examples – but there should be a more comprehensive analysis of RNAs, including head-to-head comparisons relative to other methods for identifying binding pockets.

We thank the reviewer for this observation. We would like to remind the reviewer that our benchmark set is composed of seven different RNA molecules (Tab. S1): four riboswitches, selected as representative of large and relatively stable RNAs, and three viral RNAs, more challenging targets given their smaller size and more pronounced flexibility. These two classes of molecules account for 43% of all RNAs [doi.org/10.3390/ijms24065497]. For each of these systems, we performed a total of 12 SHAMAN simulations starting from holo-like and apo, when available, conformations and we compared our results against a validation set composed of a total of 33 PDBs (with 14 “unique” pockets). These SHAMAN calculations amounted to an aggregated simulation time of ~160 μ s. Due to space limitations, in the original manuscript we provided a detailed analysis of only two of these systems, which, as the reviewer mentioned, were FMN and TAR. We have now considerably expanded our analysis and reported detailed results for all the 7 systems / 12 SHAMAN simulations. This required us to add a total of 5 pages (Supplementary Analysis) and 5 figures (Fig. S8-S12) to Supplementary Information. We believe that the detailed analysis of these systems demonstrates very clearly how SHAMAN is able to account for the flexible nature of RNA molecules when identifying small molecule interaction hotspots. From the analysis of the riboswitches, it is indeed evident how SHAMAN can accurately predict alternative binding modes after local structural rearrangements. Notably, the analysis of the viral RNAs highlighted how accounting for RNA flexibility is crucial to identify binding pockets formed upon large structural rearrangements. In this sense, the architecture of SHAMAN enables the detection of interacting hotspots that are difficult or invisible to other approaches based on static structures, like SiteMap and BiteNet (see the new section “Comparison with other tools”).

We would like to thank the reviewer again for prompting us to perform this analysis, which we believe shows even more clearly the strengths of our approach.

I think the work is important – but the claim by the authors that “Our work constitutes to date the most advanced computational pipeline for binding site identification in dynamic RNA structural ensembles, thus providing crucial information for structure-based rational design of novel compounds targeting RNA” is an overstatement not supported by the data – in my view, even if data backed the claim, the advance does not warrant publication in a general readership journal such as Nat Comm.

We agree with the reviewer that probably our statement was excessively strong. We therefore tone it down in the revised version of the manuscript, as follows:

“Our work constitutes an advanced computational pipeline for binding site identification in dynamic RNA structural ensembles, thus providing crucial information for structure-based rational design of novel compounds targeting RNA”

We have also tone down a statement in the discussion to:

“SHAMAN emerges as one of the most advanced physics-based approaches for binding site identification in RNA structural ensembles.”

However, now supported by the extended analysis of all our systems and the comparison with state-of-the-art software for binding site detection in RNA molecules, we strongly believe that our manuscript will be of interest to the general readership of Nat Comm, and we are glad that the majority of reviewers agree with us.

Reviewer #2 (Remarks to the Author):

The study by Panei et al. introduces SHAMAN, a novel computational method aimed at addressing the challenges associated with identifying small-molecule binding sites within RNA structures. In contrast to existing tools that rely on static structures, SHAMAN utilizes atomistic molecular dynamics to explore the dynamic conformational landscape of RNA while concurrently identifying potential binding pockets through parallel probes and enhanced sampling techniques, enabling the on-the-fly determination of binding affinities. The method underwent successful benchmarking against a variety of RNA structures, encompassing both rigid and flexible RNAs, with remarkably positive outcomes. SHAMAN consistently and accurately identified experimentally confirmed binding sites, ranking them among the top candidates in all cases. The paper is a significant contribution to the field, I recommend its publication after the revision of the following comments.

1. While the method's explanation is thorough, it lacks the provision of tools or algorithms for readers to apply to other systems. To address this, it is recommended that the authors share a test case, and the code including input and expected output, with the scientific community upon acceptance of the paper.

We thank the reviewer for recognizing our work as a significant contribution to the field. We agree that distribution of software and tools to perform our SHAMAN simulations is fundamental. SHAMAN simulations can be performed today using the development version of the PLUMED library (www.plumed.org), which is freely available and open source. The PLUMED input files needed to reproduce the calculations presented in our manuscript are also available on PLUMED-NEST (<https://www.plumed-nest.org/eggs/23/031>). Furthermore, for the prospective users we have created a set of python and bash scripts to facilitate the preparation of the SHAMAN input and the analysis of the results, following the step-by-step protocol described in detail in our manuscript. Our plan is to distribute these scripts with a license “free for academics / not for commercial use”. Unfortunately, this is currently not possible as the main python library that we exploit (MDAnalysis; <https://www.mdanalysis.org>) is licensed under GPLv2. The MDAnalysis developers are in process of moving to the less restrictive LGPL v3+ license (<https://www.mdanalysis.org/2022/11/07/relicensing/>), which will ultimately allow us to distribute our toolset under the license we chose.

Currently, the tools that we prepared are stored on a GitHub private repository: <https://github.com/maxbonomi/SHAMAN>. In order for the reviewer to access this private repository and possibly comment on our tools and tutorials, we have prepared a guest account:

Userid: shaman-reviewer

Password: only-for-review

We hope that the MDAnalysis licensing issues will be solved soon so that we will be able to distribute these tools to the entire community.

2. One of the paper's key strengths lies in its ability to study flexible RNA systems and the importance it places on integrating RNA dynamics with ligand sampling for binding site identification. However, these claims lack empirical support. Is the treatment of RNA flexibility important? To address that I suggest that the authors rigidify at least one flexible RNA system and repeat the study and compare the binding behavior with the flexible counterpart and experimental data. This way, we can assess whether flexibility indeed enhances prediction accuracy or not.

We thank the reviewer for raising this important point. To address this issue, we performed an additional simulation of HIV-1 TAR, which is one of the most challenging targets in our benchmark set due to its inherent flexibility. To assess the importance of accounting for flexibility in realistic drug discovery situations, we initiated this new simulation from an apo conformation of HIV-1 TAR (PDB 1anr) following our SHAMAN protocol but adding harmonic restraints (with intensity of 400 kJ/mol/nm²) to the RNA backbone atoms. With this setup, we performed a 1- μ s production run for a total aggregated simulation time (across mother and probe simulations) of 13 μ s.

As expected, during this run, HIV-1 TAR explored a single structural cluster in which we identified 23 SHAMAPs. As a term of comparison, we remind the reviewer that the HIV-1 apo simulation previously reported in our manuscript explored 7 relevant clusters and identified a total of 81 SHAMAPs. Notably, this new restrained simulation was not able to identify all the 5 distinct pockets (Tab. 2), which instead were correctly detected by SHAMAN with low values of $\Delta\Delta G$ (Tab. 1). These results clearly indicate that accounting for RNA flexibility indeed enhances prediction accuracy.

Tab. 1 Results reported in our manuscript

HIV-1 TAR RNA	$\Delta\Delta G$ [kJ/mol]	rank	best match	distance [Å]
pocket 1 PDB 1arj	1.0	23	09-PIRZ	3.9
pocket 2 PDB 1lvj	0.2	8	04-DMEE	2.9
pocket 3 PDB 1uts	0.2	8	04-DMEE	6.3
pocket 4 PDB 1uui, 1uud	1.0	23	09-PIRZ	3.8
pocket 5 PDB 218h	0.1	5	12-MAMY	3.8

Tab. 2 Results from our new restrained simulation

HIV-1 TAR RNA	$\Delta\Delta G$ [kJ/mol]	rank	best match	distance [Å]
pocket 1 PDB 1arj	-	-	-	-
pocket 2 PDB 1lvj	0.13	3	11-MEPY	0.8
pocket 3 PDB 1uts	-	-	-	-
pocket 4 PDB 1uui, 1uud	0.13	3	03-BENF	6.67
pocket 5 PDB 218h	-	-	-	-

3. The paper highlights differences in results obtained from simulations initiated with apo and holo conformations, which raises concerns regarding sampling completeness. To address this issue, it is proposed that the authors perform extended simulations for a single system to determine whether the two starting conformations converge to the same $\Delta\Delta G$ value. This analysis would help clarify the necessary simulation time for achieving thermodynamic convergence for cases where only holo-like structure is at hand.

We thank the reviewer for this comment. We are aware that our 1-microsecond long MD simulations cannot provide an exhaustive sampling of the conformational landscape of RNA molecules. Our goal for this first iteration of the method is to account for RNA flexibility in the proximity of the starting conformations and to assess whether simulations initiated from apo structures were able to capture conformations of the binding pocket resembling the holo states. We have demonstrated in all systems studied that indeed this is the case.

To address the concern of the reviewer regarding the convergence of the probe binding free energy calculations, we:

- Reported for all the binding sites identified by our probes the statistical error in the probe binding free energy in a Supplementary File called shamaps.xlsx.
- We performed two additional 2.0 μs -long simulations of HIV-1 TAR, starting from holo-like (PDB 1uts) and apo (PDB 1anr) conformations for a total aggregated simulation time of 52 μs
- We calculated the $\Delta\Delta G$ of the SHAMAPs overlapping with the experimental pocket as a function of simulation time, for apo and holo-like simulations. The results are summarized in the following figure.

Convergence of $\Delta\Delta G$ calculation. **A.** Scatter plot of the $\Delta\Delta G$ of the SHAMAP with best overlap with HIV-1 TAR pocket (validating SHAMAP) in PDB 1uts as a function of simulation time. **B.** From top to bottom, tables reporting rank, $\Delta\Delta G$, and validation distance (Eq. 10) of validating SHAMAPs for holo-like (left) and apo (right) simulations.

Remarkably, in both holo-like and apo simulations the experimental binding pocket was already detected after 0.5 μs (Figure B). In the holo-like case, already for this short simulation time the SHAMAP that identifies the ligand position in the experimental structure is among the top scored ones (red stars, A). In the apo simulation, it is instead necessary to wait until 1.0 μs to identify the experimental pocket as a top scored SHAMAP (cyan dots, A). For longer simulation, the value of $\Delta\Delta\text{G}$ can be considered the same within the statistical error. This analysis suggests that 1.0 μs is the minimum time necessary to obtain consistent identification of experimentally determined binding pockets and to converge the calculation of the SHAMAPs $\Delta\Delta\text{G}$ when starting from apo conformations.

4. Additionally, readers benefit from a systematic analysis on the crucial parameters influencing calculation accuracy, such as simulation time, number of probe replicas, number of clusters, metadynamics bias, etc. The authors are encouraged to select a test system and explore these parameters to determine their impact on prediction accuracy. Including these results in the supplementary information would help the readers to design their setups accurately.

We thank the reviewer for raising this important point. In response to this concern, we focused on two 2.0 μs simulation of HIV-1 TAR, starting from holo-like (PDB 1uts) and apo (PDB 1anr) conformations and we performed a systematic analysis of the following parameters:

- simulation time, varied between 0.5 and 2.0 microseconds (μs) in steps of 0.5 μs
- the cutoff for the RMSD *gromos* clustering of the RNA conformations, between 2.5 and 3.5 \AA in steps of 0.5 \AA
- the spacing of the grid for the free energy calculations, between 1.0 and 2.0 \AA in steps of 0.5 \AA

All the other parameters were kept constant. Regarding metadynamics, given the long-standing experience of the group and, more generally, of the metadynamics community, we do not believe that further parameter optimization is required. The Gaussian width is traditionally chosen based on the fluctuations of the collective variables in a short unbiased MD simulation, while the Gaussian deposition rate as well as the initial height and bias factor are set to 1 ps, 1.2 kJoule/mol and 10 based on the type of process under study and the nature of free-energy barriers that need to be crossed.

In our analysis we assessed the effect of our choice of parameters on two descriptors that are crucial for SHAMAN accuracy: the $\Delta\Delta\text{G}$ of the SHAMAPs that identify the experimental binding site in PDB 1uts and the distance between the SHAMAP and the experimental pocket ('validation distance', Eq. 10 in Materials and Methods).

Effect of the choice of input parameters on SHAMAN accuracy. Scatter plot of $\Delta\Delta G$ (top panels) and validation distance (Eq. 10, bottom panels) in simulations initiated from holo-like (red) and apo (blue) conformations upon variation of: **A)** simulation time, **B)** clustering cutoff, and **C)** grid spacing in probe binding free energy calculations. When one input parameter is varied, the other two are kept at their reference value used in the SHAMAN simulations reported in our manuscript ($T = 1 \mu\text{s}$, RMSD cutoff = 3.0 \AA , grid spacing = 1.0 \AA). The data reported in this figure refer to the identification of the binding pocket in the HIV-1 TAR PDB structure 1uts.

Based on this analysis, we can make the following observations.

- 1) **Simulation time.** As the simulation time increases, the SHAMAP corresponding to the experimental pocket tends to be among the most probable interacting sites (**A**, top panel) when starting our simulation from both holo-like and apo conformations. Most importantly, the estimates of $\Delta\Delta G$ from the two simulations appear to be converged and consistent given the statistical errors at play. These results support the choice we made in our original manuscript of a total simulation time of $1 \mu\text{s}$.
- 2) **RMSD clustering cutoff.** Our analysis indicates that varying the RMSD cutoff of the clustering algorithm in the range 2.5 \AA to 3.5 \AA does not significantly impact the value of $\Delta\Delta G$ of the SHAMAP that identifies the experimental binding site (**B**, top panel). The validation distance is also not undergoing significant changes upon variation of this parameter (**B**, lower panel). These results support our original choice of using a clustering cutoff equal to 3.0 \AA .

- 3) **Grid spacing.** As expected, the size of the 3D space grid directly impacts the probe binding free energy calculations. As the grid size increases, we expect the calculation of the probability of the probe to be bound to RNA and to be part of a specific pocket to be overestimated. This is a discretization issue related to assigning the position of the probe atoms to a given grid voxel. These results indicate that the grid size should be at most 1.5 Å, which is consistent with our original choice of 1.0 Å.

We reported the results of our analysis in the revised version of Supplementary Information as Fig. S13.

5. Finally, while the method's success is commendable, a comparative analysis with existing methods is essential to assess its performance comprehensively. The authors need to conduct benchmarking against other methods and present a comparison to help readers make informed decisions about adopting this technique.

We thank the reviewer for raising this fundamental point. The reviewer is absolutely right, and we apologize for this major oversight on our side.

Despite the fact that several binding site detection tools are available for proteins, there is only a limited number of approaches that can be applied to RNA molecules. In the following, we report a list of software selected as the best performing representatives of different classes of methods, i.e. grid-based, machine learning, deep learning, and regression models.

- *SiteMap* [<https://doi.org/10.1021/ci800324m>], which is part of the Schrodinger suite, uses physics-based energy calculations to predict the most probable interaction hotspots from the structure of the target. SiteMap is a state-of-the-art tool for protein binding site detection and its latest release has been optimized for nucleic acids;
- *BiteNet* [<https://doi.org/10.1093/nargab/lqab111>] uses a convolutional neural network trained on known RNA-ligand structures deposited in the PDB to predict small-molecules binding sites. *BiteNet* has been shown to be more accurate in binding site identification than four other popular tools: *RSite* [10.1038/srep09179], *RSite2* [<https://doi.org/10.1038/srep19016>], *RBind* [<https://doi.org/10.1093/bioinformatics/bty345>], and *RNA site* [10.1093/bioinformatics/btaa1092]. For this reason, we decided not to explicitly test *RSite*, *RSite2*, *RBind*, and *RNA site*.
- *RBinds* [<https://doi.org/10.1016%2Fj.csbj.2020.10.043>] is built on *RBind* and constructs a graph network from the RNA structure to predict the binding regions from the closeness and degree scores between nodes. *RBinds* has also been shown to be more accurate in binding site identification than *RSite* and *RSite2*;
- *RLBind* [<https://doi.org/10.1093/bib/bbac486>] uses a deep convolutional neural network-based model to predict RNA-ligand binding sites. RLBind has been shown to be more accurate than *RSite*, *RSite2*, *RBind* and *RNA site*. However, this tool was not compared against BiteNet in the original publication. We tested the RLBind pipeline, which is

available via a GitHub repository (<https://github.com/KailiWang1/RLBind>), and we found it not functional. Our execution was stopped at step 4.2, which was supposed to be carried out by a web server that is currently not reachable (<http://consurf.tau.ac.il>). Being the comparison impossible, the tool has been omitted from the analysis.

We compared all these methods with SHAMAN based on their ability to correctly predict the RNA nucleotides interacting with small molecules in experimentally determined structures. A detailed description of how this comparison was performed and the definition of the metrics used is reported in the revised version of our manuscript (Materials and Methods). A section named “Comparison with other tools” that illustrates the results of our analysis has been added along with the new Fig. 4. We report below the text added to the “Results” section of the main manuscript about the comparison with other tools:

“Comparison with other tools

We compared SHAMAN with three state-of-the-art computational tools for small-molecule binding site prediction on RNA molecules: SiteMap⁴⁹, BiteNet⁵⁰, and RBind^{51,52}. For all the systems in our benchmark set, we tested the ability of these tools to correctly predict the RNA nucleotides interacting with small molecules in experimentally determined structures (Materials and Methods). First, we determined the quality of the predictions obtained from holo-like conformations using only the corresponding experimental holo structure as ground truth (Tab. S1, red column). SHAMAN and BiteNet outperformed SiteMap and RBind (Fig. 4A) in terms of Matthews Correlation Coefficient (MCC score), a comprehensive measure of predictive quality for binary classifiers (Materials and Methods). The low MCC scores of SiteMap and RBind were mostly due to their low accuracy and precision. While the quality of the predictions obtained with SHAMAN and BiteNet was comparable, the precision of our approach was more variable across our benchmark set, with a tendency to overestimate the number of interacting nucleotides. Given that SHAMAN accounts for the flexibility of the RNA target, we hypothesized that this was the result of the prediction of alternative binding pockets not present in the single holo structure used as ground truth. To verify this hypothesis, we assessed the quality of predictions by considering as ground truth for each system the set of interacting nucleotides in all the experimental binding sites of our validation set (Tab. S3 and S4, Materials and Methods). With this definition, SHAMAN precision and overall MCC score improved (Fig. 4B), in support of our hypothesis. Finally, to simulate a common drug discovery scenario in which only the structure of the apo state is available, we tested the quality of the predictions obtained from apo conformations (Tab. S1, cyan column). In this case, the quality of SHAMAN predictions was superior to BiteNet (Fig. 4C) as our approach was able to identify with high accuracy and precision the correct set of interacting nucleotides in all the reference experimental structures. These results clearly indicate that prediction tools that do not account for the flexibility of the RNA target are not able to predict binding sites formed upon local or global structural rearrangements.”

Reviewer #3 (Remarks to the Author):

The paper by Panei et al. reports on an innovative approach to RNA-based computational drug design, representing a noteworthy contribution to the burgeoning field of RNA therapeutics. Employing enhanced sampling simulations along with a fragment-driven exploration of RNA pockets, the authors present a promising methodology. The new approach was then applied to riboswitches and flexible viral RNAs, addressing a critical gap in RNA-based drug discovery: identifying druggable pockets for subsequent docking and free energy simulations. Given its potential significance, the paper should be considered for publication in Nature Communications. However, several substantial and minor points warrant attention and resolution before final acceptance.

1. Comparison with Existing Tools: The authors should comprehensively compare their approach and established geometrical and structural pocket-finding tools. Combining their method with existing tools may yield even richer insights and more robust results. While fragments can help in druggability, the present methodology is very sensitive to fragment chemistry.

We thank the reviewer for raising this fundamental point. The reviewer is absolutely right, and we apologize for this major oversight on our side.

Despite the fact that several binding site detection tools are available for proteins, there is only a limited number of approaches that can be applied to RNA molecules. In the following, we report a list of software selected as the best performing representatives of different classes of methods, i.e. grid-based, machine learning, deep learning, and regression models.

- *SiteMap* [<https://doi.org/10.1021/ci800324m>], which is part of the Schrodinger suite, uses physics-based energy calculations to predict the most probable interaction hotspots from the structure of the target. SiteMap is a state-of-the-art tool for protein binding site detection and its latest release has been optimized for nucleic acids;
- *BiteNet* [<https://doi.org/10.1093/nargab/lqab111>] uses a convolutional neural network trained on known RNA-ligand structures deposited in the PDB to predict small-molecules binding sites. *BiteNet* has been shown to be more accurate in binding site identification than four other popular tools: *RSite* [10.1038/srep09179], *RSite2* [<https://doi.org/10.1038/srep19016>], *RBind* [<https://doi.org/10.1093/bioinformatics/bty345>], and *RNASite* [10.1093/bioinformatics/btaa1092]. For this reason, we decided not to explicitly test *RSite*, *RSite2*, *RBind*, and *RNASite*.
- *RBinds* [<https://doi.org/10.1016%2Fj.csbj.2020.10.043>] is built on *RBind* and constructs a graph network from the RNA structure to predict the binding regions from the closeness and degree scores between nodes. *RBinds* has also been shown to be more accurate in binding site identification than *RSite* and *RSite2*;
- *RLBind* [<https://doi.org/10.1093/bib/bbac486>] uses a deep convolutional neural network-based model to predict RNA-ligand binding sites. *RLBind* has been shown to be more accurate than *RSite*, *RSite2*, *RBind* and *RNASite*. However, this tool was not compared

against BiteNet in the original publication. We tested the RLBind pipeline, which is available via a GitHub repository (<https://github.com/KailiWang1/RLBind>), and we found it not functional. Our execution was stopped at step 4.2, which was supposed to be carried out by a web server that is currently not reachable (<http://consurf.tau.ac.il>). Being the comparison impossible, the tool has been omitted from the analysis.

We compared all these methods with SHAMAN based on their ability to correctly predict the RNA nucleotides interacting with small molecules in experimentally determined structures. A detailed description of how this comparison was performed and the definition of the metrics used is reported in the revised version of our manuscript (Materials and Methods). A section named “Comparison with other tools” that illustrates the results of our analysis has been added along with the new Fig. 4. We report below the text added to the “Results” section of the main manuscript about the comparison with other tools:

“Comparison with other tools

We compared SHAMAN with three state-of-the-art computational tools for small-molecule binding site prediction on RNA molecules: SiteMap⁴⁹, BiteNet⁵⁰, and RBind^{51,52}. For all the systems in our benchmark set, we tested the ability of these tools to correctly predict the RNA nucleotides interacting with small molecules in experimentally determined structures (Materials and Methods). First, we determined the quality of the predictions obtained from holo-like conformations using only the corresponding experimental holo structure as ground truth (Tab. S1, red column). SHAMAN and BiteNet outperformed SiteMap and RBind (Fig. 4A) in terms of Matthews Correlation Coefficient (MCC score), a comprehensive measure of predictive quality for binary classifiers (Materials and Methods). The low MCC scores of SiteMap and RBind were mostly due to their low accuracy and precision. While the quality of the predictions obtained with SHAMAN and BiteNet was comparable, the precision of our approach was more variable across our benchmark set, with a tendency to overestimate the number of interacting nucleotides. Given that SHAMAN accounts for the flexibility of the RNA target, we hypothesized that this was the result of the prediction of alternative binding pockets not present in the single holo structure used as ground truth. To verify this hypothesis, we assessed the quality of predictions by considering as ground truth for each system the set of interacting nucleotides in all the experimental binding sites of our validation set (Tab. S3 and S4, Materials and Methods). With this definition, SHAMAN precision and overall MCC score improved (Fig. 4B), in support of our hypothesis. Finally, to simulate a common drug discovery scenario in which only the structure of the apo state is available, we tested the quality of the predictions obtained from apo conformations (Tab. S1, cyan column). In this case, the quality of SHAMAN predictions was superior to BiteNet (Fig. 4C) as our approach was able to identify with high accuracy and precision the correct set of interacting nucleotides in all the reference experimental structures. These results clearly indicate that prediction tools that do not account for the flexibility of the RNA target are not able to predict binding sites formed upon local or global structural rearrangements.”

2. Fragment Selection: The fragment selection process appears somewhat arbitrary in the current study. To significantly impact the drug discovery community, the authors should consider a more rational and chemically informed approach to fragment selection. Would you be able to comment on this point and provide a more solid justification? A table detailing the selected fragment structures would also enhance the reader's understanding. Fragment selection deserves much more detail and chemical thinking.

We thank the reviewer for their important comment. We agree that fragment selection in mixed-solvent approaches is a crucial step. In SHAMAN, the fragment selection has been done using a rational chemically informed approach, which, admittedly, was not clearly described in the main manuscript. In the revised version of the manuscript, we have replaced the pre-existing paragraph in the Materials and Methods section with the following:

“Details of the probes. The set of probes used in our protocol is composed of two subsets. First, we included 8 probes already used in the SILCS-RNA study [29], namely acetate (ACEY), benzene (BENX), dimethyl-ether (DMEE), formamide (FORM), imidazole (IMIA), methyl-ammonium (MAMY), methanol (MEOH), and propane (PRPX) (Tab. S7). These fragments had been selected in the original study as a representative set of functional groups. Second, we developed the following approach to identify fragments with higher probability to bind to RNA molecules. Two databases were used, namely HARIBOSS [20] comprising 265 experimentally validated RNA binders (<https://hariboss.pasteur.cloud>) and RBIND [24] that includes 159 RNA bioactive molecules (<https://rbind.chem.duke.edu>). In an effort to identify chemical groups that exist in both libraries, we prepared the Murcko scaffolds from the molecules derived from both databases and compared the corresponding sets. 6 Murcko scaffolds appear in both HARIBOSS and RBIND molecules (Tab. S7). From these, 5 representative scaffolds were selected for the SHAMAN simulations, namely benzene (BENX), dihydro-pyrido-pyrimidinone-imidazo-pyridine (BENF), benzothiophene (BETH), methyl-pyrimidine (MEPY), and piperazine (PIRZ). The preparation and comparison of the HARIBOSS and RBIND libraries was done using a KNIME 4.6 protocol that includes the following steps: i) molecule preparation using Epik [86] at pH 7.4, ii) conversion to canonical SMILES using RDkit v. 2022.3, iii) Murcko scaffold derivation using the RDkit Murcko Scaffolds KNIME node, iv) set comparison using the ‘Compare Ligand Sets’ node provided by Schrodinger v. 2022.3, and finally v) a fragmentation of the common scaffolds using the RECAP fragmentation method [87] (implemented as the ‘Fragments from Molecules’ node provided by Schrodinger). All probes used in the SHAMAN simulations have been prepared using the LigPrep module of Schrodinger Suite at pH 7.4. [Schrödinger Release 2023-1: LigPrep, Schrödinger, LLC, New York, NY, 2023.] BETH was intentionally modeled in a protonated state, as it appears in the origin molecules from RBind and HARIBOSS.”

We would also like to underline the fact that two tables reporting the selected fragment structures were already present in our original manuscript (Tab. S7 and S8).

3. **Selectivity and Multitarget Drug Discovery:** Given the growing concern in RNA-based drug discovery regarding selectivity, the paper should discuss whether the new methodology offers any insights or solutions. Additionally, exploring the potential for multitarget drug discovery in the realm of RNA therapeutics would be valuable.

We thank the reviewers for raising the important point of selectivity. We recognize that our current implementation of SHAMAN does not solve this fundamental issue. One of the main reasons is that the set of probes that we developed for our identification of binding sites in RNA structural ensembles are in part obtained from fragmentation of RNA binders extracted from our HARIBOSS database. We do not know whether these molecules are selective for specific RNAs in the first place nor that fragments derived from such ligands preserve their selectivity. Unfortunately, we are not aware of extensive collections of RNA ligands that are demonstrated experimentally to be selective and therefore at this stage we cannot develop a better set of probes. In our mind the problem of selectivity can be addressed during the phase of virtual screening of pockets identified by SHAMAN, where small molecules can be screened *in silico* against different RNAs and binders potentially selective for specific RNAs identified. We have added to the revised version of the manuscript (Discussion) the following paragraph to comment on the issue of selectivity:

“One of the growing concerns with rational drug discovery approaches for RNA targeting is selectivity. Although in the present study we apply SHAMAN to RNA molecules with low sequence identity, one could consider employing our protocol to examine the uniqueness of a binding site in one target against a set of undesirable targets close in sequence (antitargets). In the case where a binding site is located in the same area across all examined RNA molecules, but it has different physico-chemical and structural properties, a cross-docking approach, i.e. docking to multiple RNAs and selecting molecules with predicted affinity for the desired target significantly higher compared to the others, can be used to identify potentially selective compounds.”

4. **Combination with Virtual Screening:** Although the text mentions combining the approach with virtual screening protocols, the specifics of this integration still need to be clarified. Clarifying how these two methods can be effectively combined would enhance the methodology's practical applicability. One may expect the virtual screening outcomes to be very much biased by the chemical structure of the original fragments.

We thank the reviewer for this very important comment. To explore how to integrate our SHAMAN pipeline in a virtual screening campaign, we have initiated a collaboration with the Forli group at Scripps Research (USA), who develops the popular Autodock docking software. The idea is to use the probe free energy maps computed with SHAMAN to guide docking of small molecules. The output of our pipeline is a set of density maps in mrc format that can be directly integrated in Autodock using the framework that the Forli group has been developing to aid docking with cryo-electron microscopy density maps. Furthermore, the Forli group has been developing an optimized scoring function to rank small molecule binders to RNAs, which will also be part of our virtual screening pipeline. Since our efforts are based on preliminary results not yet published

from another research group, we would prefer not to disclose our plans in the current manuscript. We hope that the reviewer understands this delicate issue.

5. **Fragment Library Screening:** As an extension of point 4, the authors might consider the feasibility of screening a small library of fragments and ranking them using their new method. This could provide a practical demonstration of its utility and the general applicability of the technique. In brief, if fragments chemically out of those used for pocket detection are systematically discarded, the methodology may be weak in general applicability.

We thank the reviewer for the interesting suggestion. It is important to note that the fragments used in mixed-solvent approaches like ours are not necessarily the most successful fragments in a fragment-based drug discovery (FBDD) campaign and vice versa. For example, as fragment hits are most often low-affinity binders, they need to be synthetically linked, grown, or merged once identified. As a consequence, in FBDD, the library to be used often consists of molecules with multiple functional groups as growth vectors. This is not a requirement in SHAMAN. The current set of SHAMAN probes includes: *i*) probes rationally selected as the common fragmented scaffolds obtained from two libraries of RNA specific small molecules, i.e. HARIBOSS and RBIND; *ii*) a diverse set of fragments used by the SILCS-RNA approach. No further restrictions were applied (see question 2 above). Nevertheless, we plan in the future to perform a FBDD approach for a specific target and validate SHAMAN using the fragment hits. At the moment, we believe that this is beyond the scope of the present study.

6. **Free Energy Estimations:** The paper needs more information on the errors associated with the reported free energy estimations. Precision in free energy values, including their associated uncertainties, should be carefully considered (how many digits)?

We thank the reviewer for raising this important point. In the original version of our manuscript, we reported in Fig. S7 the distribution of statistical errors in the probe binding free energy across all the sites discovered by SHAMAN. To complement this information, in the revised version of the manuscript we are providing a Supplementary Excel file (shamaps.xlsx) in which each pocket found by our approach is annotated with the statistical error in the probe binding free energies.

7. **Experimental Validation:** In the future, the authors should prioritize experimental validation of their predictions through techniques such as surface plasmon resonance (SPR) or microscale thermophoresis (MST), ITC and so on. Such validation is crucial for establishing the credibility and practical utility of the new methodology.

We thank the reviewer for raising this important point. Naturally, validation of our predictions in terms of location of binding pockets and affinity of the probes/ligands is crucial. For this paper, the location of the binding pockets identified by SHAMAN was experimentally validated by construction as our benchmark set was composed of systems for which RNA/small molecules

structures were already determined experimentally. We are now applying SHAMAN to a specific RNA target, namely MALAT1. In a preliminary hit finding campaign of medium-throughput screening (MTS), two molecules sharing the same scaffold were identified as active without, however, any knowledge of their binding site. Our SHAMAN approach followed by docking calculations has identified one specific binding site as the most promising. Our collaborators are in the process of experimentally validating our predictions.

We are excited to hear the authors' response to these concerns and hope to see the further development and validation of their promising approach in the near future.

We thank the reviewer for acknowledging the importance of our work.

REVIEWERS' COMMENTS

Reviewer #2 (Remarks to the Author):

All my comments and concerns has been addressed. I recommend acceptance of the study.

Reviewer #3 (Remarks to the Author):

The paper by Panei et al. addresses one emerging field in drug design and discovery: targeting RNA with small molecules. They developed a new computational tool immediately available to the community via PLUMED. Hence, other researchers can benefit from the code, run simulations, and discover novel lead candidates. The manuscript was reviewed, and the Authors correctly addressed all the points raised by the previous referees. It is the opinion of the present reviewer that the manuscript deserves rapid publication in Nature Communications. Only one minor point for the Discussion may deserve consideration by the Authors: how RNA-based drug discovery, in the arena of the (crowded) small-molecule drug discovery, can benefit from chemical diversity and provide new ideas for potential drugs. The following review article can help the Authors better grasp the potential of small-molecule drug discovery in RNA and elaborate more on the possible chemical diversity for modulating RNA (Computational drug discovery under RNA times. QRB Discovery, 1-21 5, 2022).

Reviewer #3 (Remarks to the Author)

The paper by Panei et al. addresses one emerging field in drug design and discovery: targeting RNA with small molecules. They developed a new computational tool immediately available to the community via PLUMED. Hence, other researchers can benefit from the code, run simulations, and discover novel lead candidates. The manuscript was reviewed, and the Authors correctly addressed all the points raised by the previous referees. It is the opinion of the present reviewer that the manuscript deserves rapid publication in Nature Communications. Only one minor point for the Discussion may deserve consideration by the Authors: how RNA-based drug discovery, in the arena of the (crowded) small-molecule drug discovery, can benefit from chemical diversity and provide new ideas for potential drugs. The following review article can help the Authors better grasp the potential of small-molecule drug discovery in RNA and elaborate more on the possible chemical diversity for modulating RNA (Computational drug discovery under RNA times. QRB Discovery, 1-21 5, 2022).

We thank the reviewers for raising this point. We have added the suggested reference to the revised version of the manuscript.